# FourierNets enable the design of highly non-local optical encoders for computational imaging

**Diptodip Deb**[1]     **Zhenfei Jiao**[1,2]     **Ruth R. Sims**[1,3]     **Alex B. Chen**[1,4]

**Michael Broxton**[5,†]     **Misha B. Ahrens**[1,†]     **Kaspar Podgorski**[1,†]     **Srinivas C. Turaga**[1,†]

[1] HHMI Janelia Research Campus, [2] Wuhan National Laboratory for Optoelectronics, HUST
[3] Institut de la Vision, Sorbonne Université, CNRS, [4] Harvard University, [5] Stanford University,
`{debd,simsr,ahrensm,podgorskik,turagas}@janelia.hhmi.org`
`zfjiao@hust.edu.cn`
`abchen@g.harvard.edu`
`broxton@alumni.stanford.edu`
[†] equal contribution

## Abstract

Differentiable simulations of optical systems can be combined with deep learning-based reconstruction networks to enable high performance computational imaging via end-to-end (E2E) optimization of both the optical encoder and the deep decoder. This has enabled imaging applications such as 3D localization microscopy, depth estimation, and lensless photography via the optimization of local optical encoders. More challenging computational imaging applications, such as 3D snapshot microscopy which compresses 3D volumes into single 2D images, require a highly non-local optical encoder. We show that existing deep network decoders have a locality bias which prevents the optimization of such highly non-local optical encoders. We address this with a decoder based on a shallow neural network architecture using global kernel Fourier convolutional neural networks (Fourier-Nets). We show that FourierNets surpass existing deep network based decoders at reconstructing photographs captured by the highly non-local DiffuserCam optical encoder. Further, we show that FourierNets enable E2E optimization of highly non-local optical encoders for 3D snapshot microscopy. By combining Fourier-Nets with a large-scale multi-GPU differentiable optical simulation, we are able to optimize non-local optical encoders $170\times$ to $7372\times$ larger than prior state of the art, and demonstrate the potential for ROI-type specific optical encoding with a programmable microscope.

## 1   Introduction

Modern computational optics relies on the end-to-end (E2E) optimization of the optical system (optical encoder) and the computational image reconstruction algorithm (computational decoder). This has been enabled by the development of differentiable physics-based simulations of optical encoders, paired with deep network based decoders. This E2E approach has been successfully applied to the estimation of depth from defocus [1, 2, 3, 4], lensless photography [5], and particle localization microscopy [6, 7]. Despite this wide success, only the space of local optical encoders has been explored in this fashion due to the computational expense of physics-based simulation of highly non-local optical encoders [3]. This is particularly the case for 3D snapshot microscopy which is well-known to require highly non-local optical encoders, and for which no E2E solutions have yet been demonstrated.

36th Conference on Neural Information Processing Systems (NeurIPS 2022).

We show that there are actually two issues blocking the E2E optimization of highly non-local optical encoders. First, the computational expense of physics-based simulation of non-local optical encoders is significantly greater, since they are orders of magnitude larger than local encoders. And second, the locality bias inherent to deep network architectures currently used for computational imaging prevents decoding from non-local solutions. In this paper, we primarily focus on developing a computational framework for engineering highly non-local optical encoders by solving both issues. We solve the first issue by developing a large-scale multi-GPU differentiable optical simulation, and the second by developing FourierNet: a new decoder network architecture based on Fourier convolutions.

In 3D snapshot microscopy, all existing methods [8, 9, 10, 11, 12, 13, 14, 15, 16, 17, 18, 5, 19] employ a highly non-local optical encoder to optically transform a 3D volume into a single 2D camera image, which is computationally decoded to reconstruct the 3D volume. Fast volumetric imaging is invaluable across biology, including for imaging neural activity throughout the whole brain of an animal [20, 21, 22]. Here, we focus on developing a 3D snapshot microscope for imaging neuronal activity across the whole brain of the larval zebrafish (*Danio rerio*) at camera rates exceeding 100 Hz. This is two orders of magnitude faster than the fastest conventional microscopes — light sheet imaging of whole brain neural activity can only achieve 0.5 Hz - 2 Hz volume rates [20]. This improvement of temporal resolution is essential for imaging with fast calcium indicators [23] and voltage indicators [24].

Newly developed programmable microscopes with up to $10^6$ free parameters, e.g. pixels on a spatial light modulator (SLM), enable the implementation of a rich space of optical encodings and present the possibility of direct optimization of snapshot microscope parameters specifically for particular ROI types and imaging tasks. While our paper primarily focuses on addressing the computational challenges of E2E optimization of highly non-local optical encoders, we also demonstrate a proof-of-concept that an SLM-based programmable microscope can implement such an engineered non-local optical encoder, paving the way to implementing multiple ROI-type and task specific optical encoders in a single physical microscope. In contrast, 3D snapshot imaging has classically been performed using fixed optical encoders based on microlens arrays [8, 9, 10, 11, 12, 13, 14, 15], pseudorandom diffusers/masks [16, 17], and designed or optimized diffusers/masks [18, 5, 19], with some of these methods using deep learning for reconstruction [11, 15].

**Problem statement** We define an optical encoder as $\mathbf{M_\phi}$ parameterized by $\phi$ and a computational decoder as $\mathbf{R_\theta}$ parameterized by $\theta$. Optical encoders can usually be modeled as linear transfer functions implemented by the optics. In special cases, such as for the optical systems explored in this paper, they can be represented as a linear convolution filter called a "point spread function" with coefficients computed by a wave optics simulation, dependent on the physical parameters $\phi$ of the optical system (Appendix A.1). We wish to develop a reconstruction network architecture for decoding non-local encodings in 2D images $\mathbf{c}$ produced by $\mathbf{M_\phi}$ and also to enable the end-to-end optimization of $\phi$ to produce non-local encodings. We'll consider two applications to investigate these optimization problems: (1) lensless photography (Figure 2), where we optimize only the reconstruction network $\mathbf{R_\theta}$ to reconstruct images of natural scenes from camera images captured by the DiffuserCam [25], and (2) 3D snapshot microscopy using an SLM-based programmable microscope (Figure 1A), where we perform E2E optimization of both the reconstruction network $\mathbf{R_\theta}$ and the optical encoding $\mathbf{M_\phi}$ based on the SLM parameters in order to image 3D volumes $\mathbf{v}$ and reconstruct 3D volumes $\hat{\mathbf{v}}$. While we focus on solving the computational challenges involved in this E2E optimization of a large highly non-local optical encoder, we also demonstrate a proof-of-concept implementation of our optimized optical encoder using an SLM-based programmable microscope.

## 1.1 Our contributions

1. We developed a large-scale, parallel, multi-GPU differentiable wave optics simulation of a programmable microscope, based on a $4f$ optical model with a phase mask ($\phi$) implemented using a spatial light modulator (SLM), described further in Appendix A.1. SLMs ($\phi$) can have over $10^6$ optimizable parameters, which we can feasibly simulate and optimize to produce PSFs with $170\times$ to $7372\times$ more unique voxels than previous attempts at deep learning PSF optimization using single GPUs [6, 3] (Appendix A.5).

2. We collected a large dataset of high resolution 3D confocal volumes of zebrafish larvae for the purpose of ROI-type specific end-to-end optimization of optical encoders.

3. We introduce an efficient FourierNet reconstruction network architecture for decoding from non-local optical encoders using very large global convolutions implemented via Fourier convolutions.

4. We show that our networks outperform the state-of-the-art deep decoders for DiffuserCam based lensless photography [25] and for 3D snapshot microscopy.

5. Our method enables, for the first time, direct end-to-end optimization of highly non-local optical encoders in the space of spatial light modulator (SLM) pixels with over $10^6$ parameters. In simulation, we demonstrate the potential for significant improvements in imaging resulting from ROI-type specific optimization of optical encoders.

## 1.2 Prior work

Neural network architectures for computational imaging have all used convolution layers with small filters. End-to-end optimization of optical encoders have largely been performed with UNet-based architectures [1, 2, 3, 4, 5, 7], with one method using a ResNet-based architecture [6]. Such optimization has always led to local optical encodings. End-to-end optimization has never been attempted for large-field of view 3D snapshot microscopy due to the difficulty of simulating and reconstructing from non-local encoders. However, small filter convolutional deep networks have been used in a non-end-to-end manner to reconstruct volumes from 3D snapshot microscopes designed using microlens arrays [15, 26, 27]. One recent hybrid approach combines a UNet with a differentiable approximate inverse method (Wiener filter) to handle non-local spatially varying (non-convolutional) optical encoders [27]. For photography and MRI, another hybrid approach of deep learning combined with unrolling iterations of traditional deconvolution algorithms provides the benefits of fast amortized optimization and higher quality reconstructions due to learning of structural priors [28, 29, 25, 26]. We note that these hybrid approaches typically require measurement of the optical encoder of the system, whereas our method does not and can learn to produce high quality reconstructions using only pairs of ground truth and system images.

In our work, we demonstrate that convolution layers with large filters implemented efficiently in the Fourier domain enable the end-to-end learning of highly non-local optical encoders for 3D snapshot microscopy. Large convolution filters have been shown to be helpful for other computer vision applications such as semantic segmentation and salient object detection [30, 31], and the Fourier domain parameterization of small filters has been described previously [32].

A pioneering strategy in 3D snapshot microscopy is light field microscopy [8], which employs a microlens array at the microscope's image plane to create subimages encoding both the amplitude and phase of light [8, 33]. A variety of microlens-array-based light field microscopes have been used to perform whole brain imaging [8, 11, 14, 22, 13, 12, 9, 34, 19]. [19] optimizes the placement of microlenses, but not in an end-to-end manner. Despite variation in design, microlens-based microscopes have, to various degrees, four main limitations that can be improved: 1) blocking or scattering light between microlenses, causing light inefficiency, 2) not making use of all pixels on the camera to encode a 3D volume, leading to inefficient compression and suboptimal reconstructions $\hat{\mathbf{v}}$, 3) aliasing at some planes with generally nonuniform axial resolution, and 4) a fixed optical encoding scheme.

An alternative to using microlenses is to implement a coded detection strategy using a phase mask or diffuser to spread light broadly across the camera sensor [16, 18, 35, 5, 10, 17, 19]. The designed phase masks can be implemented either by manufacturing a custom optical element or using a programmable SLM [2, 3, 4, 5, 19, 6]. Using a programmable element allows different microscope parameters to be used for different sample and ROI types.

## 2 Methods

We show our network architecture and an overview of autoencoder training both the microscope parameters $\phi$ and reconstruction network parameters $\theta$ in Figure 1A. The programmable microscope is simulated by a differentiable implementation of a wave-optics model of light propagation. We have selected a programmable microscope design based on pupil-plane phase modulation with a programmable spatial light modulator, for which imaging is well-approximated by a computationally-efficient convolution [36]. A detailed description of our simulation is provided in Appendix A.1. For lensless photography, there is no optical simulation because the images have been collected on a real camera.

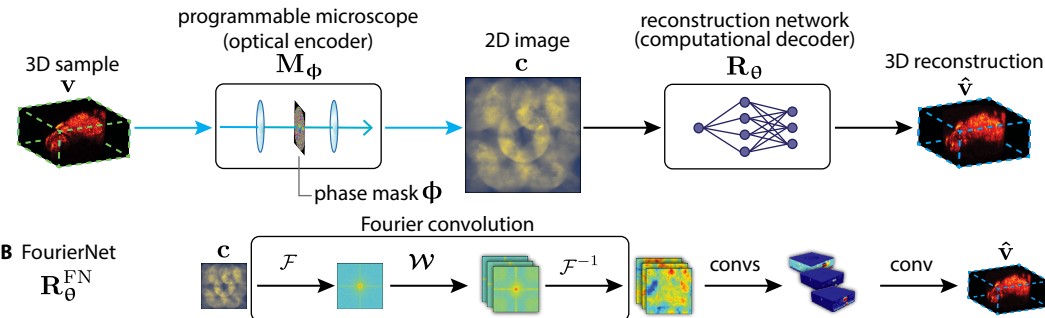

Figure 1: Overview of our problem setup and our proposed network architectures. Top row (**A**) shows the problem of 3D snapshot microscopy, where we computationally reconstruct a 3D volume from a 2D image. Bottom row (**B**) shows our proposed FourierNet architecture, which includes a **Fourier convolution** layer that enables efficient computation of global features.

## 2.1 FourierNet for decoding non-local optical encoders

Because images created by optical encoders can potentially encode signals from the incoming light field to any location in the camera image in a non-local manner, it is essential that a reconstruction network have global context. Existing multi-scale architectures such as the UNet [37] can achieve global context, but at the expense of many computation layers with small convolution kernels which we speculate to have a local information bias that is inappropriate for computational optics. In this paper, we introduce relatively shallow architectures for computational imaging, which rely on convolution layers with very large filters, implemented efficiently in the Fourier domain [32].

**FourierNet** We propose a simple three layer convolutional network architecture with very large global convolutions at the very first layer, followed by two standard local convolutional layers (Figure 1B). We define a global convolution as a convolution with kernel size equal to the input image. Such a convolution achieves global context in a single step but is computationally expensive. Global convolutions are implemented more efficiently in the Fourier domain, yielding a speed up of two orders of magnitude over direct convolution. Due to the use of Fourier convolutions to enable global context, we call our architecture FourierNet. In contrast to a typical UNet which can contain many tens of convolution layers, the FourierNet is only three layers deep, which requires backpropagation through fewer layers compared to a typical UNet with the same receptive field.

**Fourier domain convolutions** It is well-known that large kernel convolutions can be implemented more efficiently in the Fourier domain [38, 39, 40]. A naive implementation of global convolution requires $\mathcal{O}(N^2)$ operations, where $N$ is the number of pixels in both the image and the kernel. An alternative global convolution implementation is to Fourier transform ($\mathcal{F}$) the input $\mathbf{x}$ and convolution kernel $\mathbf{w}$, perform element-wise multiplication in Fourier space, and finally inverse Fourier transform ($\mathcal{F}^{-1}$), requiring only $\mathcal{O}(N \log N)$ operations [39, 40]. Following [32], we store and optimize the weights $\mathcal{W}$ in Fourier space. This over-parameterization costs $8\times$ the memory of an equivalent real valued large filter but saves the computational cost of Fourier transforming the real-valued weights (Appendix A.6). Thus a Fourier convolution is defined:

$$\mathbf{Re}\{\mathcal{F}^{-1}\{\mathcal{W} \odot \mathcal{F}\{\mathbf{x}\}\}\} \tag{1}$$

For image and kernel sizes of $256 \times 256$ pixels, our implementation leads to nearly $500\times$ speedup: a standard PyTorch convolution takes 2860 ms, while a Fourier convolution takes 5.92 ms on a TITAN X. We also show how we can naturally extend our Fourier convolutions and FourierNet to a multiscale version using cropping in the Fourier domain in Appendix A.2.

## 2.2 Physics-based autoencoder for simultaneous engineering of microscope encoder and reconstruction network decoder

We describe the imaging process as the following transformation from the 3D intensity volume $\mathbf{v}$ to the 2D image formed on the camera $\mathbf{c}$:

$$\boldsymbol{\mu_c} = \mathbf{M_\phi}(\mathbf{v}) \tag{2}$$

$$\mathbf{c} = \max\left([\boldsymbol{\mu_c} + \sqrt{\boldsymbol{\mu_c}}\epsilon], 0\right), \epsilon \sim \mathcal{N}(0, 1) \tag{3}$$

where $\mathbf{M}_{\boldsymbol{\phi}}$ denotes the microscope parameterized by a 2D phase mask, $\boldsymbol{\phi}$. This phase mask $\boldsymbol{\phi}$ describes the 3D-to-2D encoding of this microscope model completely. A Poisson distribution with mean rate $\boldsymbol{\mu_c}$ describes the physics of photon detection at the camera, but sampling from this distribution is not differentiable. We approximate the noise distribution with a rectified Gaussian. We include details on $\mathbf{M}_{\boldsymbol{\phi}}$ in Appendix A.1 [36]. Jointly training reconstruction networks and microscope parameters involves image simulation, reconstruction, then gradient backpropagation to update the reconstruction network and microscope parameters. Our parallelization strategy enables optimization of phase masks with millions of parameters in a feasible amount of time and memory per GPU. This also enables us to produce PSFs with multiple orders of magnitude more unique voxels than previous attempts (Appendix A.4). Details on parallelization, planewise reconstruction networks, and planewise sparse gradients are provided in Appendix A.4, Figure 12.

We use the normalized mean squared error (NMSE) as the basis for our loss function. Since we care more about the high spatial frequency content of the image which contains mostly cells, and less about the low spatial frequencies which contain mostly background, we implemented a two part loss function. $L_{\mathrm{HNMSE}}$ measures the NMSE between the high pass filtered volume and high pass filtered reconstruction. This is combined with the unfiltered NMSE $L_{\mathrm{NMSE}}$ between the original volume and its reconstruction. Formally, our loss function $L(\mathbf{v}, \hat{\mathbf{v}})$ for all snapshot microscopy reconstruction problems is defined as:

$$L(\mathbf{v}, \hat{\mathbf{v}}) = L_{\mathrm{HNMSE}}(\mathbf{v}, \hat{\mathbf{v}}) + \beta L_{\mathrm{NMSE}}(\mathbf{v}, \hat{\mathbf{v}}) \tag{4}$$

$$L_{\mathrm{HNMSE}}(\mathbf{v}, \hat{\mathbf{v}}) = \frac{\mathbb{E}\left[(H(\mathbf{v}) - H(\hat{\mathbf{v}}))^2\right]}{\mathbb{E}(H(\mathbf{v})^2)}, L_{\mathrm{NMSE}}(\mathbf{v}, \hat{\mathbf{v}}) = \frac{\mathbb{E}\left[(\mathbf{v} - \hat{\mathbf{v}})^2\right]}{\mathbb{E}(\mathbf{v}^2)} \tag{5}$$

where $H(\cdot)$ denotes high pass filtering and $\mathbb{E}(\cdot)$ denotes the mean over pixels and sample volumes. Both loss terms are normalized as shown to reduce variance in $L$, which can otherwise cause large magnitude fluctuations based on the brightness variation across training volumes resulting in training instability. For our experiments, we set the weight $\beta$ for the $L_{\mathrm{NMSE}}$ term to 0.1.

## 3 Results

**DiffuserCam Lensless Mirflickr Dataset** We test reconstruction performance on experimental computational photography data[1] from [25] (Figure 2). This is a dataset constructed by displaying RGB color natural images from the MIRFlickr dataset on a monitor and then capturing diffused images by the DiffuserCam lensless camera. The dataset contains 24,000 pairs of DiffuserCam and ground truth images. The goal of the dataset is to learn to reconstruct the ground truth images from the diffused images. As in [25], we train on 23,000 paired diffused and ground truth images, and test on 999 held-out pairs of images.

**Larval Zebrafish Snapshot Microscopy Dataset** We show all our results for snapshot microscopy using our simulation of a snapshot microscope, and our engineered optical encoders have not been experimentally tested for reconstruction performance on a programmable microscope. We simulate snapshot imaging using high resolution confocal imaging volumes of zebrafish. These are volumes of transgenic larval zebrafish whole brains expressing the nuclear-restricted GCaMP6 calcium indicator in all neurons. These images are representative of brain-wide activity imaging. We train on 58 different zebrafish volumes (which we augment heavily) and test on 10 held-out volumes. For all experiments, we downsample the high resolution confocal data to (1.0 µm z, 1.625 µm y, 1.625 µm x). We created 4 datasets from these scanned volumes corresponding to imaging different fields of view or regions of interest (ROIs) for our experiments. Full specifications for these datasets are in Table 6 (Appendix A.4). For Figure 3, we restrict the field of view to (200 µm z, 416 µm y, 416 µm x) with a tall cylinder cutout of diameter 193 µm and height 200 µm and image with $256 \times 256$ pixels on the simulated camera sensor. Figure 4 and Table 3 show our larger experiments with $512 \times 512$ simulated camera pixels, with a field of view of (250 µm z, 832 µm y, 832 µm x).

### 3.1 FourierNets outperform state-of-the-art for reconstructing natural images captured by DiffuserCam lensless camera

We compare our FourierNet architecture to the best learned method from [25] using unrolled ADMM and a denoising UNet, as well as to a vanilla UNet from [25]. Architecture details are in Appendix

---

[1]Publicly available: https://waller-lab.github.io/LenslessLearning/dataset.html

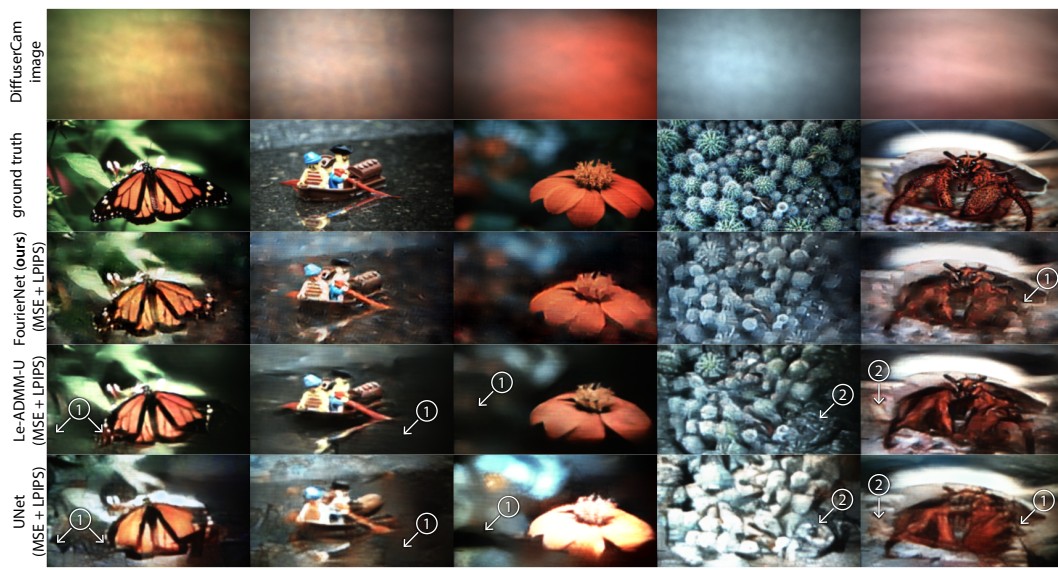

Figure 2: Comparisons of our method (third row) to state-of-the-art learned reconstruction methods on lensless diffused images of natural scenes. Regions labeled ① indicate missing details, either resolution or textures in backgrounds. Regions labeled ② indicate hallucinated textures. Note that the previous state-of-the-art solution (fourth row) [25] exhibits both issues more often compared to our method.

Table 1: Quality of natural image reconstruction on the DiffuserCam Lensless Mirflickr Dataset (mean ± s.e.m., $n = 999$). Superscripts denote loss function: [1] MSE, [2] MSE+LPIPS.

| Method | MSE ↓ ($\times 10^{-2}$) | LPIPS ↓ | MS-SSIM ↑ | PSNR ↑ | Time ↓ (ms) |
|---|---|---|---|---|---|
| FourierNet[1] | **0.39 ± 0.007** | 0.20 ± 0.00 | **0.882 ± 0.001** | **24.8 ± 0.09** | 35.54 |
| FourierNet[2] | 0.54 ± 0.010 | **0.16 ± 0.00** | 0.868 ± 0.001 | 23.4 ± 0.09 | 35.54 |
| Le-ADMM-U[2] [25] | 0.75 ± 0.021 | 0.19 ± 0.00 | 0.865 ± 0.002 | 22.1 ± 0.09 | 48.59 |
| UNet[2] [25] | 1.68 ± 0.060 | 0.24 ± 0.00 | 0.818 ± 0.002 | 19.2 ± 0.11 | **06.97** |

A.6, A.7. We can see that FourierNet visually outperforms the methods from [25] in Figure 2, and quantitatively in Table 1 across all quality metrics. The Le-ADMM-U and UNet results in Table 1 were reported by [25] using a combined MSE + LPIPS loss. Unlike [25], we find that training FourierNets with the MSE loss alone provides reconstructions visually similar to the ground truth as shown in Figure 13 (Appendix A.7). Timings in Table 1 are for only the forward pass on a single TITAN Xp GPU.

### 3.2 FourierNets outperform UNets for engineering non-local optical encoders

Table 2: Quality of reconstructed volumes after optimizing microscope parameters to image zebrafish on $256 \times 256$ pixel camera (mean ± s.e.m., $n = 10$)

| Microscope | Reconstruction | $L_{\text{HNMSE}}$ ↓ | MS-SSIM ↑ | PSNR ↑ | Time ↓ (s) |
|---|---|---|---|---|---|
| FourierNet2D | FourierNet3D | **0.6093 ± 0.0209** | **0.955 ± 0.004** | **34.89 ± 0.88** | **0.38** |
| FourierNet2D | UNet3D | 0.7298 ± 0.0151 | 0.923 ± 0.008 | 30.16 ± 0.94 | 0.96 |
| Wiener + UNet | Wiener + UNet | 0.7223 ± 0.0179 | **0.957 ± 0.003** | 34.49 ± 0.91 | 0.73 |
| UNet2D | UNet3D | 0.7109 ± 0.0161 | 0.913 ± 0.009 | 29.17 ± 1.13 | 0.96 |

We compare optimizing microscope parameters $\boldsymbol{\phi}$ with three neural networks: 1) using our FourierNet with 2D convolutions (FourierNet2D), 2) using a vanilla UNet with 2D convolutions (UNet2D), and 3) using a Wiener deconvolution as an approximate inverse combined with a refining UNet, as

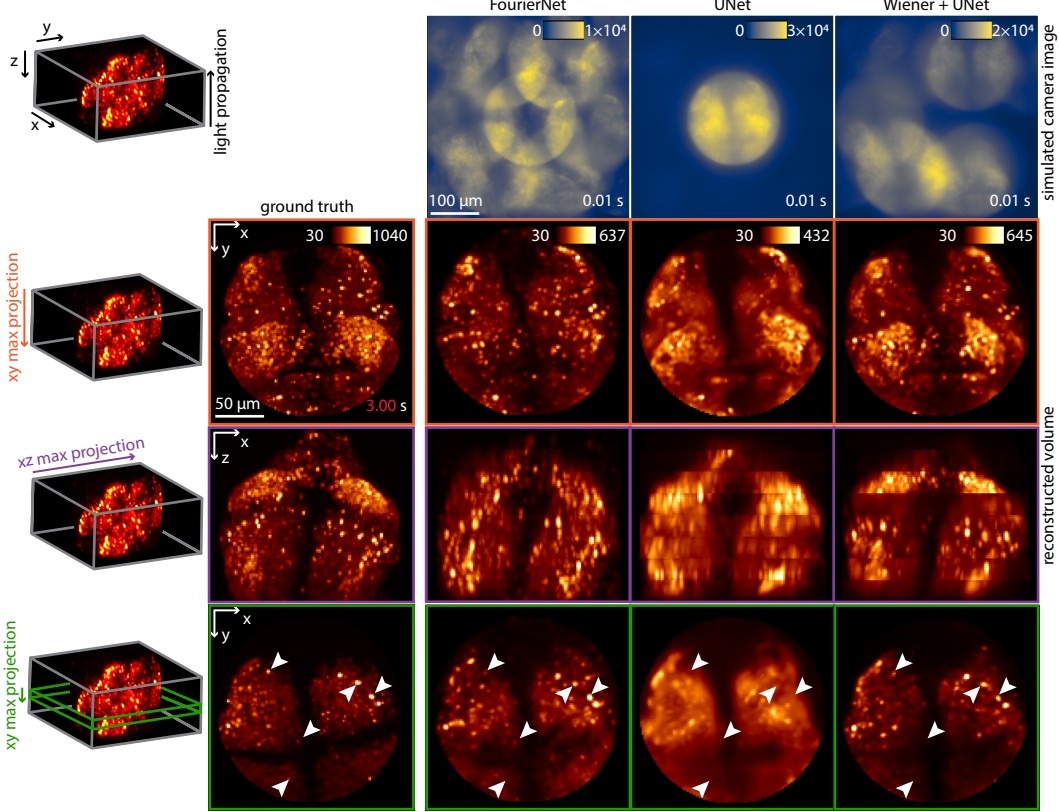

Figure 3: Comparing simulated camera images (0.01 second expected acquisition time) and corresponding reconstructions of a volume captured using our FourierNet (left) versus UNet (middle) and Wiener + UNet [3] (right) optimized microscopes. Top row shows simulated $256 \times 256$ pixel camera images; bottom right of camera image shows approximate acquisition time (given a reasonable number of simulated photons per pixel, i.e. SNR). Ground truth has no corresponding camera image, because the 3D volume is imaged directly via slow high resolution confocal microscopy (3 second acquisition time). Colored arrows in left column show projection axis for each row. White arrows show individual neurons clearly visible for FourierNet, but not for UNet. Wiener + UNet [3] has detail in some planes, but not consistently throughout the whole volume, and also uses fewer camera pixels.

in [3]. Training is a two stage process in which a plane-wise reconstruction network with fewer parameters and 2D convolutions is used during optimization of $\phi$, then once $\phi$ is fixed a more powerful reconstruction network with 3D convolutions is used, except for the Wiener + UNet [3] method which is trained in a single stage as in [3]. Table 2 shows the type of network used for the first stage to train the optical encoder $\mathbf{M}_\phi$ in the first column, and the type of network used in the second stage for training the reconstruction network only (given a fixed $\phi$) in the second column. We find this scheme achieves better reconstruction quality because the reconstruction network does not need to constantly adapt to a changing optical encoding (Appendix A.6). We train these microscopes and reconstruction networks on ROIs which are tall cylindrical cutouts of zebrafish with diameter 193 μm and height 200 μm. Sample volumes are imaged on a camera with $256 \times 256$ pixels (Figure 3). FourierNet2D has 2 convolution layers (with 99.8% of its kernel parameters in the initial Fourier convolution layer), while UNet2D has 32 convolution layers (kernel parameters approximately uniformly distributed per layer). UNet2D was designed to have a global receptive field. FourierNet2D and UNet2D both have $\sim 4 \times 10^7$ parameters; FourierNet3D has $\sim 6 \times 10^7$ parameters vs. $\sim 10^8$ for UNet3D. The Wiener + UNet method [3] has $\sim 8 \times 10^7$ parameters. Architecture details are in Appendix A.6, A.8.

Simulated camera images (Figure 3) show that the UNet microscope does not make sufficient use of camera pixels, producing only a single view of the volume. We speculate this is due to a local information prior in the small kernels of UNets. Adding a Wiener deconvolution as an approximate

Table 3: ROI specific microscope parameter optimization for 3 types of zebrafish volumes (mean PSNR (top), MS-SSIM (bottom) $\pm$ s.e.m., $n = 10$). Green shows regions of interest.

| Tested on | Microscope parameters optimized for | | |
| | Type A | Type B | Type C |
|---|---|---|---|
| Type A | **49.76 $\pm$ 1.35** | 46.03 $\pm$ 1.33 | 42.67 $\pm$ 1.14 |
| | **0.998 $\pm$ 0.000** | 0.996 $\pm$ 0.001 | 0.992 $\pm$ 0.002 |
| Type B | 35.56 $\pm$ 1.41 | **37.34 $\pm$ 0.96** | 35.38 $\pm$ 1.16 |
| | 0.965 $\pm$ 0.004 | **0.972 $\pm$ 0.003** | 0.967 $\pm$ 0.003 |
| Type C | 30.87 $\pm$ 1.15 | 31.48 $\pm$ 0.93 | **33.79 $\pm$ 0.90** |
| | 0.912 $\pm$ 0.007 | 0.920 $\pm$ 0.006 | **0.935 $\pm$ 0.006** |

inverse before a UNet also does not result in a microscope that makes full use of camera pixels, but is better than the UNet alone. The FourierNet microscope uses more camera pixels and performs better than the UNet microscope for reconstruction (Figure 3, quantified in Table 2). Timings in Table 2 are for only the forward pass on a single TITAN Xp GPU; one training iteration on 8 GPUs takes ~0.4 seconds for FourierNet3D, ~0.8 seconds for Wiener + UNet, and ~0.8 seconds for UNet3D (Appendix A.6). Both reconstruction networks must reconstruct from images that have a compressed encoding of 3D information, but the FourierNet2D is clearly more effective than the UNet2D at optimizing this encoding.

### 3.3 FourierNets outperform UNets for 3D snapshot microscopy volume reconstruction

We can determine which architecture is better for volume reconstruction by choosing fixed microscope parameters and varying the architecture, except for the Wiener + UNet method [3] which is trained in one stage. In Table 2, we compare results using a FourierNet with 3D convolutions (FourierNet3D) and a vanilla UNet with 3D convolutions (UNet3D). UNet3D was also designed to have a global receptive field. Architecture details are in Appendix A.6, A.8.

Reconstruction results in Table 2 compare normalized MSE $L_{\text{HNMSE}}$ between the high pass filtered volume and high pass filtered reconstruction, the multiscale structural similarity $\text{MS} - \text{SSIM}$ between the true volume and its reconstruction, and finally the peak signal-to-noise ratio PSNR. We also visualize reconstruction results for a volume in the head of a zebrafish in Figure 3. The UNet3D reconstruction networks (using either microscope) fall significantly behind the FourierNet3D reconstruction network in all metrics, despite their global receptive field. The Wiener + UNet [3] network achieves similar $\text{MS} - \text{SSIM}$ and PSNR as the FourierNet but a worse $L_{\text{HNMSE}}$ due to the inconsistent detail, though the reconstructions are slightly better than the UNet3D for certain regions of the volume.

### 3.4 Engineered optical encoders can be optimized for a region of interest

To explore the effect of ROI size on optimized $\phi$ and the resulting reconstruction performance, we optimized $\phi$s for three different regions of interest: 1) Type A, ROIs with a short cylinder cutout of 386 μm diameter and 25 μm height, 2) Type B, ROIs with a tall cylinder cutout of 386 μm diameter and 250 μm height, and 3) Type C, ROIs without any cutout of dimension (250 μm z $\times$ 832 μm y $\times$ 832 μm x) (Table 3). All ROI types were imaged with $512 \times 512$ pixels on the simulated camera. We then tested reconstruction performance on all combinations of optimized $\phi$ and ROI type, as shown in Table 3 and visualized in Figure 4 for Type B. We include architecture details in Appendix A.6, A.9.

We see in Table 3 that for all types, highest performance is achieved using the phase mask optimized for that particular type. These results show that there is potentially a large benefit to optimizing type-specific optical encoders, which can be easily implemented in a programmable microscope.

### 3.5 Engineered optical encoders can be implemented on a programmable microscope

We demonstrate that the optical encoders engineered by simulations can be implemented on real hardware using a prototype programmable microscope. Our engineered phase masks were displayed

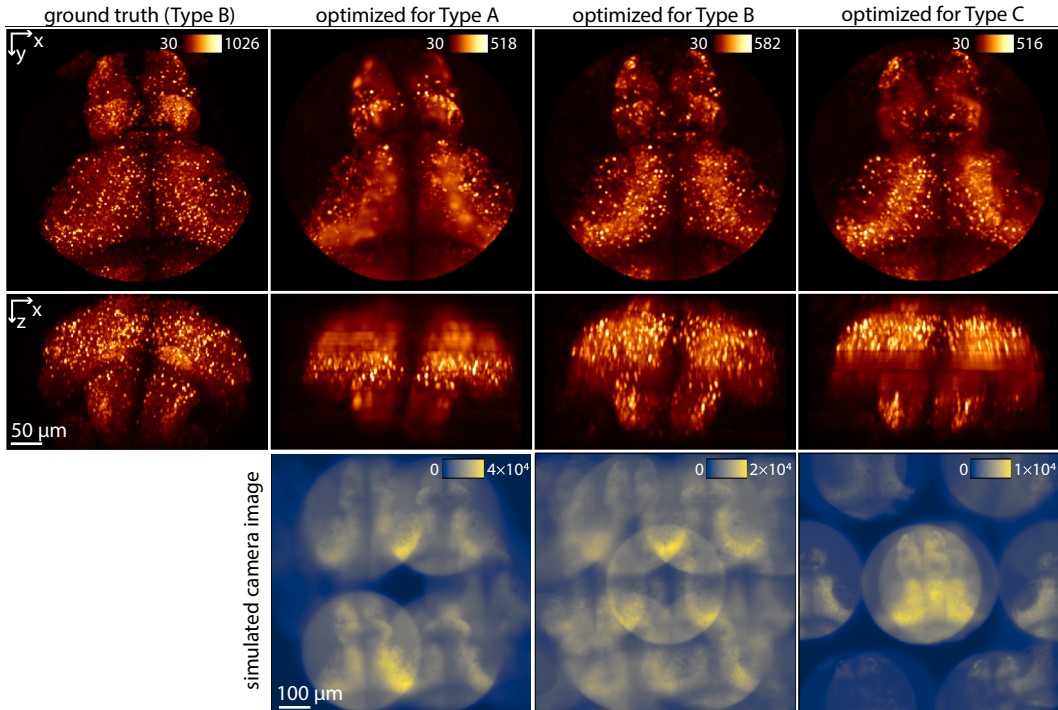

Figure 4: Reconstructed volumes resulting from imaging Type B ROIs by microscopes optimized for Types A, B, and C. Imaging Type B ROIs with microscope parameters optimized for Type B ROIs yields the best reconstructions. Top to bottom: xy max projection, xz max projection, simulated $512 \times 512$ camera image.

on a spatial light modulator (SLM) located in the pupil plane. We observe a good qualitative match between the features of the simulated and experimental optical encoders (point spread functions) in Figure 5, technical details in Appendix A.10. There is also a good qualitative match between the simulated camera images generated by both optical encoders. These results demonstrate the potential for programmable microscopy with spatial light modulators.

We also note that our optimized optical encoders result in pencil-like elements, qualitatively similar to lenslet-based approaches [19, 10, 9]. However as we show in our supplement, our optimized phase masks (Figures 11) appear qualitatively different from lenslet-based phase masks, with different regions of the pupil contributing to the same pencil. This suggests a qualitatively different, and perhaps more light efficient, mechanism for generating high resolution projections of the volume along different axes.

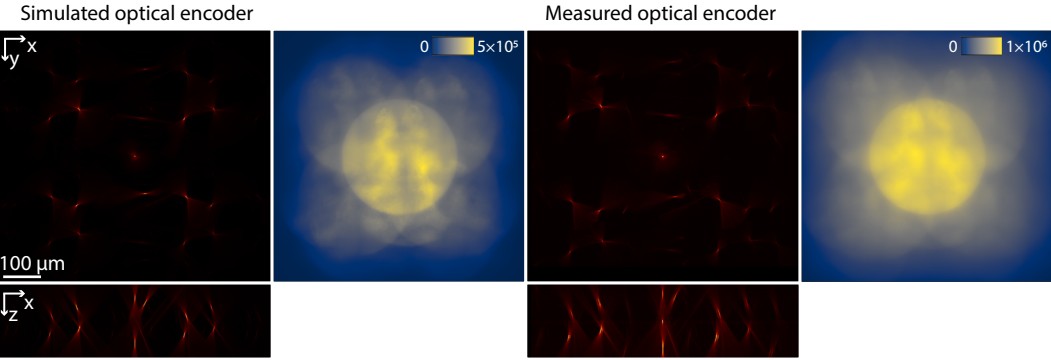

Figure 5: Simulated versus measured optical encoders (point spread functions) with corresponding simulated camera images using an example from our Type B zebrafish dataset. Zoom in for best viewing.

# 4 Discussion

**Summary** We have presented the FourierNet architecture as a deep network based decoder for computational imaging with highly non-local optical encoders. We demonstrated the superiority of FourierNets for lensless photography on the DiffuserCam dataset, and also for the end-to-end optimization of highly non-local optical encoders (where UNets fail) for 3D snapshot microscopy. FourierNets are many orders of magnitude faster than traditional iterative reconstruction algorithms [25] and generate higher quality reconstructions by learning image priors. Generally, our global Fourier-domain convolution architecture could be applicable to other problems where global integration of features is necessary, though we have focused on computational imaging applications where physics-based optical encoders induce global mixing of information. Our work solves two important computational problems preventing E2E optimization of large highly non-local optical encoders: the computational complexity of simulating large non-local encoders, and the effective decoding from such encoders. Our contributions are primarily computational, however we also demonstrate the potential for implementing 3D snapshot microscopy with an E2E optimized non-local optical encoder using an SLM-based programmable microscope. And in simulation, we demonstrate the potential for ROI-specific optimization of optical encoders for 3D snapshot microscopy, which can be efficiently implemented in a programmable microscope.

**Limitations** While we have fully demonstrated that our framework now enables computational E2E optimization of large non-local optical encoders, we do not focus on their hardware implementation. The specific claims regarding the potential benefits of ROI-specific optimization of optical encoders are only evaluated in simulation, and have only been evaluated for whole brain larval zebrafish data.

**Reproducibility** We train on 8 Quadro RTX 8000 GPUs for the largest experiments, and have described our pre-processing, training, and testing procedures in Appendix A.4, A.6, A.8. We have made our simulation software, training scripts, and our datasets available at https://github.com/TuragaLab/snapshotscope.

**Broader Impact** End-to-end optimization of optics can improve image quality but also enable multiplexed imaging not currently possible with standard optics. Our PyTorch-based optical modeling library as well as our neural network architectures are publicly available and could enable new experiments in neuroscience via whole brain imaging at an order of magnitude greater temporal resolution. Our training procedure does require long optimization periods with many GPUs, which poses a carbon footprint and barrier to usage compared to conventional microscopy.

## Acknowledgement

We would like to thank William Bishop, Roman Vaxenburg, Gert-Jan Both, Janne Lappalainen, Nathan Klapoetke, Richard Xu, Lu Mi, Sridhama Prakhya, and Jinyao Yan for invaluable feedback and discussions. We thank Howard Hughes Medical Institute and a Simons Foundation grant (Simons Collaboration on the Global Brain, 542943SPI, Ahrens) for their funding. We also thank Huazhong University of Science and Technology and the China Scholarship Council for supporting the work of Zhenfei Jiao. We additionally thank the Janelia Visiting Scientist Program for supporting the work of Ruth Sims.

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
