# A Appendix

## A.1 Forward simulation of programmable 3D snapshot microscope

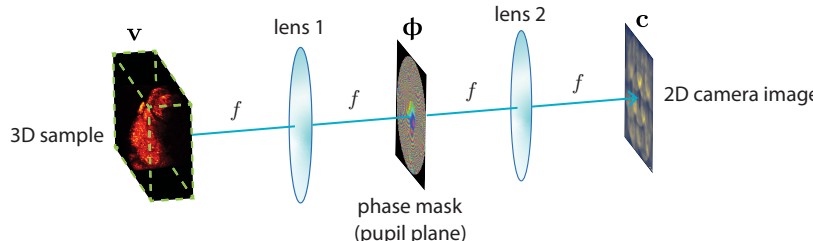

Figure 6: Diagram of a $4f$ optical model that is the basis for our simulated microscope $\mathbf{M}_{\boldsymbol{\phi}}$, showing the Fourier plane in which we have the programmable and trainable 2D phase mask $\boldsymbol{\phi}$.

Here we describe our wave optics simulation of the microscope $\mathbf{M}_{\boldsymbol{\phi}}$, which we model as a $4f$ system [36]. The $4f$ optical system consists of two lenses: the first spaced one focal length after the object plane and the second spaced one focal length before the camera image plane (Figure 6). In between these two lenses, we can place a phase mask to manipulate the light field before passing through the second lens and forming an image on the camera sensor.

We are concerned here with fluorescence microscopy, meaning that the sources of light that we image are individual fluorescent molecules, which we can model as point emitters. Because these molecules emit incoherent light, the camera sensor in effect sums the contributions of each point emitter. In order to model such an imaging system, we first need to address modeling a single point emitter's image on the camera.

We can analytically calculate the complex-valued light field one focal length after the first lens (which we call the pupil plane) due to a point source centered at some plane $z$ (where $z$ is a distance from the object plane $z = 0$). If the point source were centered ($x = 0, y = 0$) in the object focal plane $z = 0$, we would have a plane wave at the pupil plane, but for the more general case of a point source at an arbitrary plane $z$ relative to the object plane $z = 0$, we can analytically calculate the complex-valued light field entering the pupil plane:

$$u_{\text{point}}(\mathbf{k}; z) = \exp\left[i2\pi z\sqrt{\left(\frac{n}{\lambda}\right)^2 - ||\mathbf{k}||_2^2}\right] \tag{6}$$

where $u_{\text{point}}$ is the incoming light field entering the pupil due to a point source centered in the plane at $z$, $\mathbf{k} \in \mathbb{R}^2$ denotes frequency space coordinates of the light field in the pupil plane, $n$ is the refractive index, and $\lambda$ is the wavelength of light [36].

In this pupil plane, we can then apply a phase mask $\boldsymbol{\phi}$ to the light field, which is modeled as a multiplication of $u_{\text{point}}(\mathbf{k}; z)$ and $e^{i\boldsymbol{\Phi}(\mathbf{k})}$, the complex phase of the pupil function. The light field exiting the pupil is therefore described by

$$u_{\text{pupil}}(\mathbf{k}; z) = u_{\text{point}}(\mathbf{k}; z)p(\mathbf{k}) \tag{7}$$

where $p(\mathbf{k})$ is the pupil function, composed of an amplitude $a(\mathbf{k})$ and phase $\boldsymbol{\phi}(\mathbf{k})$:

$$p(\mathbf{k}) = a(\mathbf{k})e^{i\boldsymbol{\Phi}(\mathbf{k})} \tag{8}$$

$$a(\mathbf{k}) = \begin{cases} 1 & ||\mathbf{k}||_2 \leq \frac{\text{NA}}{\lambda} \\ 0 & ||\mathbf{k}||_2 > \frac{\text{NA}}{\lambda} \end{cases} \tag{9}$$

where NA is the numerical aperture of the lens [36].

The light field at the camera plane can then be described by a Fourier transform [36]:

$$\mathbf{u}_{\text{camera}} = \mathcal{F}\{\mathbf{u}_{\text{pupil}}\} \tag{10}$$

The camera measures the intensity of this complex field:

$$s(\mathbf{x}; z) = |u_{\text{camera}}(\mathbf{x}; z)|^2 \tag{11}$$

where $\mathbf{x} \in \mathbb{R}^2$ denotes spatial coordinates in the camera plane [36].

We can call this intensity $\mathbf{s}$ the point response function (PRF). If the shape of the PRF is translationally equivariant in $\mathbf{x}$, meaning that moving a point source in-plane creates the same field at the camera, shifted by the corresponding amount, then we call this PRF a point spread function (PSF), which we also refer to as an optical encoder. Note that axially translating the point source in $z$ changes the profile of the field at the camera, which allows our system to encode depth information through the PSF [35].

In order to avoid edge effects during imaging, we simulate the PSF at a larger field of view, then crop and taper the edges of the PSF:

$$\mathbf{s}_{\text{taper}} = \text{crop}[\mathbf{s}] \odot \mathbf{t} \tag{12}$$

where $\mathbf{t}$ is a taper function created by taking the sigmoid of a distance transform divided by a width factor controlling how quickly the taper goes to 0 at the edges and $\odot$ denotes elementwise multiplication. We intentionally simulate a larger field of view than the sample volume in order to avoid edge artifacts. The purpose of the $\text{crop}[\cdot]$ is to restrict the PSF to the correct field of view. The purpose of the tapering is to remove artifacts at the edges of the cropped PSF. After we compute this cropped and tapered PSF, we also downsample $\mathbf{s}_{\text{taper}}$ to the size of the data $\mathbf{v}$ in order to save memory.

Imaging is equivalent to the convolution of the incoming light field volume intensity $\mathbf{v}$ and the cropped and tapered PSF $\mathbf{s}_{\text{taper}}$ for a given plane. At the camera plane, the light field intensity is measured by the camera sensor. Therefore, we can describe the forward model as the following convolution and integral over planes:

$$\boldsymbol{\mu}_{\mathbf{c}}(\mathbf{x}) = \iint v(\boldsymbol{\tau}_{\mathbf{x}}; z) s_{\text{taper}}(\mathbf{x} - \boldsymbol{\tau}_{\mathbf{x}}; z) \, d\boldsymbol{\tau}_{\mathbf{x}} \, dz \tag{13}$$

We then model shot noise of the camera sensor to produce the final image $\mathbf{c}$, for which the appropriate model is sampling from a Poisson distribution with a mean of $\boldsymbol{\mu}_{\mathbf{c}}$ [36]:

$$\mathbf{c} \sim \text{Poisson}\left(\boldsymbol{\mu}_{\mathbf{c}}\right) \tag{14}$$

However, because we cannot use the reparameterization trick to take pathwise derivatives through the discrete Poisson distribution, we instead approximate the noise model with a rectified Gaussian distribution:

$$\epsilon \sim \mathcal{N}(0, 1) \tag{15}$$

$$\mathbf{c} \approx \max\left(\left[\boldsymbol{\mu}_{\mathbf{c}} + \sqrt{\boldsymbol{\mu}_{\mathbf{c}}}\epsilon\right], 0\right) \tag{16}$$

We now turn our attention to selecting the number of pixels used in the phase mask, i.e. the number of parameters for $\mathbf{M}_{\boldsymbol{\phi}}$. We first need to determine the pixel size for Nyquist sampling the image plane with an objective of a given NA (numerical aperture). For a given pixel size $\Delta x$, we know that in frequency space coordinates we will have a bandwidth of $\frac{1}{\Delta x}$, spanning $-\frac{1}{2\Delta x}$ to $\frac{1}{2\Delta x}$. Because we must have

$$||\mathbf{k}||_2 \leq \frac{\text{NA}}{\lambda} \tag{17}$$

we know that the Nyquist sampling pixel size is given by

$$\Delta x^* = \frac{\lambda}{2\text{NA}}. \tag{18}$$

Therefore, in the image plane, for a desired field of view $L$ we must have at least

$$N^* = \frac{L}{\Delta x^*} \tag{19}$$

pixels. The discretization in the pupil plane will be the same, which means we will need to have at least $N^*$ pixels in the pupil plane to achieve the appropriate light field in the image plane. For our settings of NA $= 0.8$, $\lambda = 0.532\mu\text{m}$, and $L = 823\mu\text{m}$, we have $\Delta x^* = 0.3325\mu\text{m}$ and $N^* = 2476$ pixels. Thus, a reasonable choice is $\Delta x = 0.325\mu\text{m}$ and $N = 2560$ pixels. Note that these simulation parameters are independent of the camera pixels; we have only determined how many pixels must

be used in the phase mask in order to ensure our PSF can occupy the full field of view. The camera sensor can sample the field at the image plane at an independent pixel size.

## A.2 Multiscale feature extraction using FourierUNets

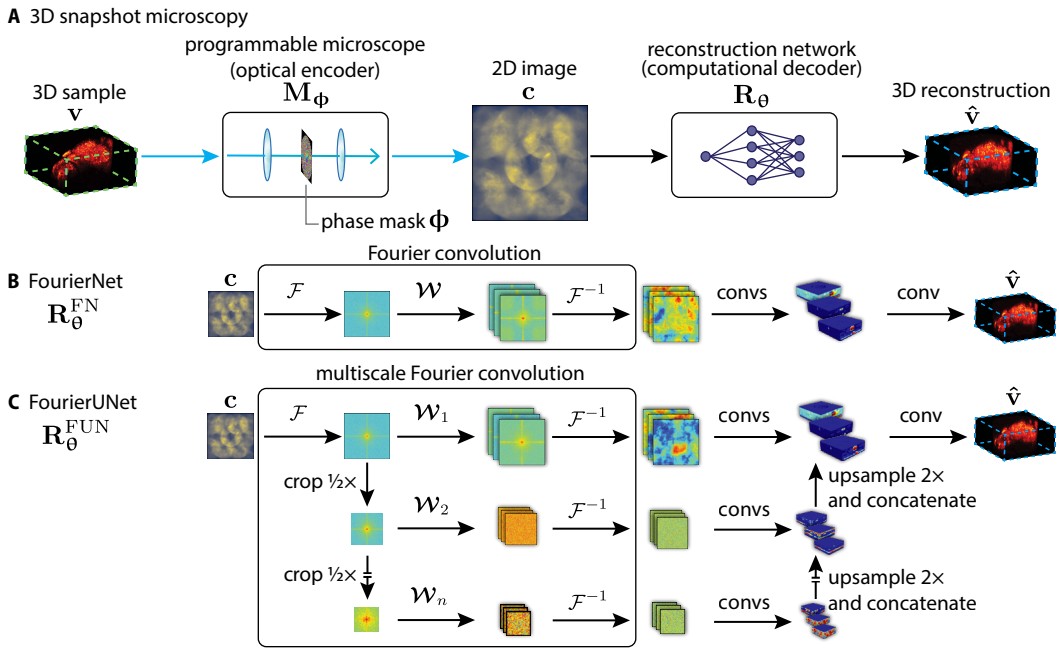

Figure 7: Overview of our problem setup and our proposed network architectures. Top row (**A**) shows the problem of 3D snapshot microscopy, where we computationally reconstruct a 3D volume from a 2D image. Middle row (**B**) shows our proposed FourierNet architecture, which includes a **Fourier convolution** layer that enables efficient computation of global features. Bottom row (**C**) shows an extension of our proposed architecture, the FourierUNet, which mimics the multiscale feature extraction of a standard UNet efficiently and with global features using a **multiscale Fourier convolution**.

**FourierUNet** We propose a multi-scale variant of the FourierNet by bringing together elements of the multi-scale UNet and the single-scale FourierNet. Here, we take advantage of the fact that down-sampling in image space corresponds to a simple cropping operation in the Fourier domain, resulting in a band-limited computation of a feature map. We efficiently implement multi-scale global Fourier convolutions (Figure 1C) to replace the encoding/"analysis" pathway of a UNet. We then use the standard decoding/"synthesis" pathway of the UNet to combine the multi-scale features into a single 3D volume reconstruction (Appendix A.6, A.8). Thus we can study whether multi-scale features or global context is more important for decoding non-local optical encoders.

**Multi-scale Fourier domain convolutions** It is well-known [38] that downsampling corresponds to cropping in the Fourier domain. Thus the Fourier convolution can be extended to efficiently produce multi-scale feature representations in one step (Figure 1C). We define our multi-scale Fourier convolution as

$$\left\{ \mathbf{Re}\{\mathcal{F}^{-1}\left\{\boldsymbol{\mathcal{W}_1} \odot \text{crop}_1\left[\mathfrak{c}\right]\right\}\}, \ldots, \mathbf{Re}\{\mathcal{F}^{-1}\left\{\boldsymbol{\mathcal{W}_n} \odot \text{crop}_n\left[\mathfrak{c}\right]\right\}\}\right\} \tag{20}$$

where subscript denotes scale level (higher subscript indicates lower spatial scale/more cropping in Fourier space) and we precompute $\mathfrak{c} := \mathcal{F}\left\{\mathbf{c}\right\}$ once.

## A.3 Global receptive field is more important than multiscale features

UNets are effective because: (1) features are computed at multiple scales and (2) large receptive fields are achieved in few layers. FourierUNets allowed us to decouple these two explanations because the receptive field is global in a single layer. We see on both our microscopy dataset (which does not have multiscale structure) and the lensless photography dataset (which does have multiscale structure) that

Table 4: Quality of natural image reconstruction on the DiffuserCam Lensless Mirflickr Dataset (mean ± s.e.m., $n = 999$). Superscripts denote loss function: [1] MSE, [2] MSE+LPIPS.

| Method | MSE ↓ ($\times 10^{-2}$) | LPIPS ↓ | MS-SSIM ↑ | PSNR ↑ | Time ↓ (ms) |
|---|---|---|---|---|---|
| FourierNet[1] | **0.39 ± 0.007** | 0.20 ± 0.00 | **0.882 ± 0.001** | **24.8 ± 0.09** | 35.54 |
| FourierNet[2] | 0.54 ± 0.010 | **0.16 ± 0.00** | 0.868 ± 0.001 | 23.4 ± 0.09 | 35.54 |
| FourierUNet[1] | 0.43 ± 0.009 | 0.22 ± 0.00 | 0.875 ± 0.001 | 24.5 ± 0.09 | 83.63 |
| FourierUNet[2] | 0.66 ± 0.012 | 0.18 ± 0.00 | 0.853 ± 0.001 | 22.6 ± 0.09 | 83.63 |
| Le-ADMM-U[2] [25] | 0.75 ± 0.021 | 0.19 ± 0.00 | 0.865 ± 0.002 | 22.1 ± 0.09 | 48.59 |
| UNet[2] [25] | 1.68 ± 0.060 | 0.24 ± 0.00 | 0.818 ± 0.002 | 19.2 ± 0.11 | **06.97** |

Table 5: Quality of reconstructed volumes after optimizing microscope parameters to image zebrafish on $256 \times 256$ pixel camera (mean ± s.e.m., $n = 10$)

| Microscope | Reconstruction | $L_{\text{HNMSE}}$ ↓ | MS-SSIM ↑ | PSNR ↑ | Time ↓ (s) |
|---|---|---|---|---|---|
| FourierNet2D | FourierNet3D | **0.6093 ± 0.0209** | **0.955 ± 0.004** | **34.89 ± 0.88** | **0.38** |
| FourierNet2D | FourierUNet3D | **0.5997 ± 0.0219** | **0.956 ± 0.003** | **34.87 ± 0.82** | 0.72 |
| Wiener + UNet | Wiener + UNet | 0.7223 ± 0.0179 | **0.957 ± 0.003** | **34.49 ± 0.91** | 0.73 |
| FourierNet2D | UNet3D | 0.7298 ± 0.0151 | 0.923 ± 0.008 | 30.16 ± 0.94 | 0.96 |
| UNet2D | UNet3D | 0.7109 ± 0.0161 | 0.913 ± 0.009 | 29.17 ± 1.13 | 0.96 |

the FourierUNet does not improve upon the FourierNet. Thus we see that it is more important for decoding from non-local optical encoders to have a global receptive field than multi-scale features.

### A.4 Training PSFs and volume reconstruction networks

Given a simulation of imaging, we can define two modes of autoencoder training: (1) jointly training the phase mask parameters $\boldsymbol{\phi}$ and weak reconstruction networks in order to learn a good optical encoder for a particular class of ROIs (i.e. samples with the same spatiotemporal statistics), and (2) training a stronger reconstruction network only with a fixed, pre-trained $\boldsymbol{\phi}$.

**Definition of terms** For both cases of training, the general framework is to simulate imaging using confocal volumes of pan-neuronal labeled larval zebrafish, reconstruct from the simulated image, then update the reconstruction network and, if desired, the microscope parameters. We will define the microscope parameters as $\boldsymbol{\phi}$ and the reconstruction network parameters as $\boldsymbol{\theta}$ for any reconstruction network $\mathbf{R}_{\boldsymbol{\theta}}(\mathbf{c})$ where $\mathbf{R}_{\boldsymbol{\theta}}$ maps 2D images to 3D volume reconstructions. For our training algorithms listed below, we also define: $\mathbf{D}$ our **dataset**, $\mathbf{v}$ a **ground truth volume**, $\hat{\mathbf{v}}$ a **reconstructed volume**, $L$ a computed **loss**, $z_s$ a list of $z$ plane indices that will be imaged/reconstructed, $\alpha_{\boldsymbol{\phi}}$ the learning rate for the microscope parameters, $\alpha_{\boldsymbol{\theta}}$ the learning rate for the reconstruction network parameters, and $\beta$ the weight of the non-high pass filtered component of the loss. When selecting a random ground truth volume, we also perform random shift, rotation, flip, and brightness augmentations.

**Microscope simulation parameters** When simulating the zebrafish imaging, we use a wavelength of 0.532 µm for all simulations. The NA of our microscope is 0.8. The refractive index $n$ is 1.33. We downsample all volumes to (1.0 µm z, 1.625 µm y, 1.625 µm x). We use a taper width of 5 for all simulations, and simulate the optical encoder (PSF) at 50% larger dimensions in x and y. The resolution of the camera (for all zebrafish datasets) is also (1.625 µm y, 1.625 µm x).

**Initialization of $\boldsymbol{\phi}$** For Type A, B, and our small $256 \times 256$ pixel experiments, we initialize $\boldsymbol{\phi}$ to produce an optical encoder (PSF) consisting of 6 pencil beams at different locations throughout the depth of the volume, with the centers of these beams arranged in a hexagonal pattern in x and y. Because our optimizations generally find optical encoders (PSFs) with many pencils, we find that initializing with such a pattern helps to converge to a more optimal optical encoder (data not shown).

For Type C, we instead initialize with a single helix spanning the depth of the volume (the "Potato Chip" from [35]), which seems to find a local minimum for $\boldsymbol{\phi}$ that produces an optical encoder with more pencils (and therefore views in the camera image).

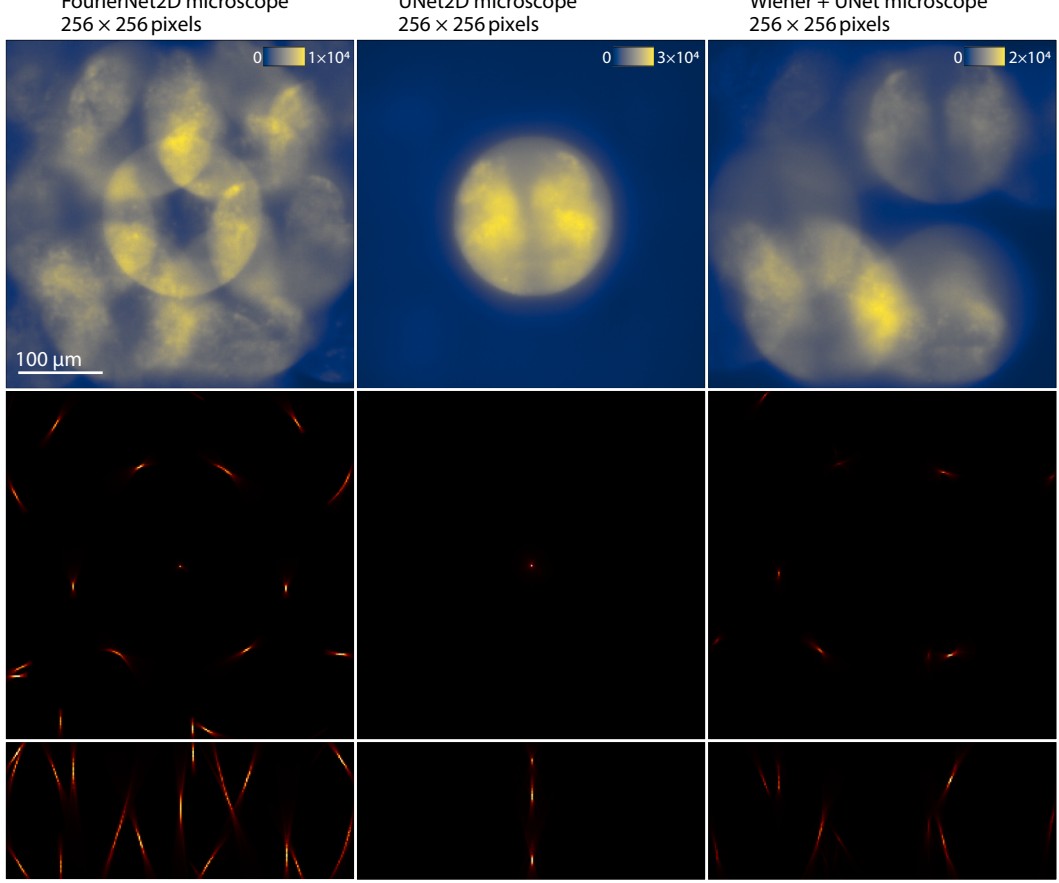

Figure 8: FourierNet successfully optimizes an optical encoder (PSF) to image and reconstruct zebrafish where UNet fails. The FourierNet learned to produce multiple pencils in its optical encoder, which create multiple views of the volume in the camera image. UNet learned only a single pencil and fails to utilize the majority of pixels in the camera image to encode views of the volume. Wiener + UNet produced an optical encoder with multiple pencils, but they do not make as optimal use of the camera pixels as the FourierNet optical encoder. Top row shows simulated camera image of a zebrafish using each optical encoder, middle row shows xy max projection of the optical encoder, and bottom row shows xz max projection of the optical encoder.

**Data settings and augmentation for zebrafish data** Using our total 58 training zebrafish volumes and 10 testing zebrafish volumes (imaged through confocal microscopy), we crop in four different ways to create four different datasets. For training volumes, we crop from random locations from each volume as a form of augmentation. For testing, we crop from the same location. Physically, these crops correspond to either placing a circular aperture before light hits the $4f$ system or changing the illumination thickness in $z$, because samples would be illuminated from the side in a real implementation of this microscope. We model these by cropping cylinders (or cubes if there is no aperture) of different diameters and heights. We show details for all types Type A, B, C in Table 6, where the diameter of the cylinder is labeled "Aperture Diameter" and the illumination thickness is labeled "Height". For our small initial experiments to compare UNets and FourierNets for optimizing phase masks, we simulated a camera with $256 \times 256$ pixels and during reconstruction each volume had 96 planes, a field of view of (200 μm z, 416 μm y, 416 μm x), and a cylindrical cutout diameter of 193 μm.

We augment our volumes during training by taking random locations from these volumes, randomly flipping the volumes in both z and y, and also randomly rotating in pitch, yaw, and roll. Most importantly, we also randomly scale the brightness of our samples and add random background levels which serve to adjust the signal-to-noise ratio (SNR) of the resulting simulated images. The

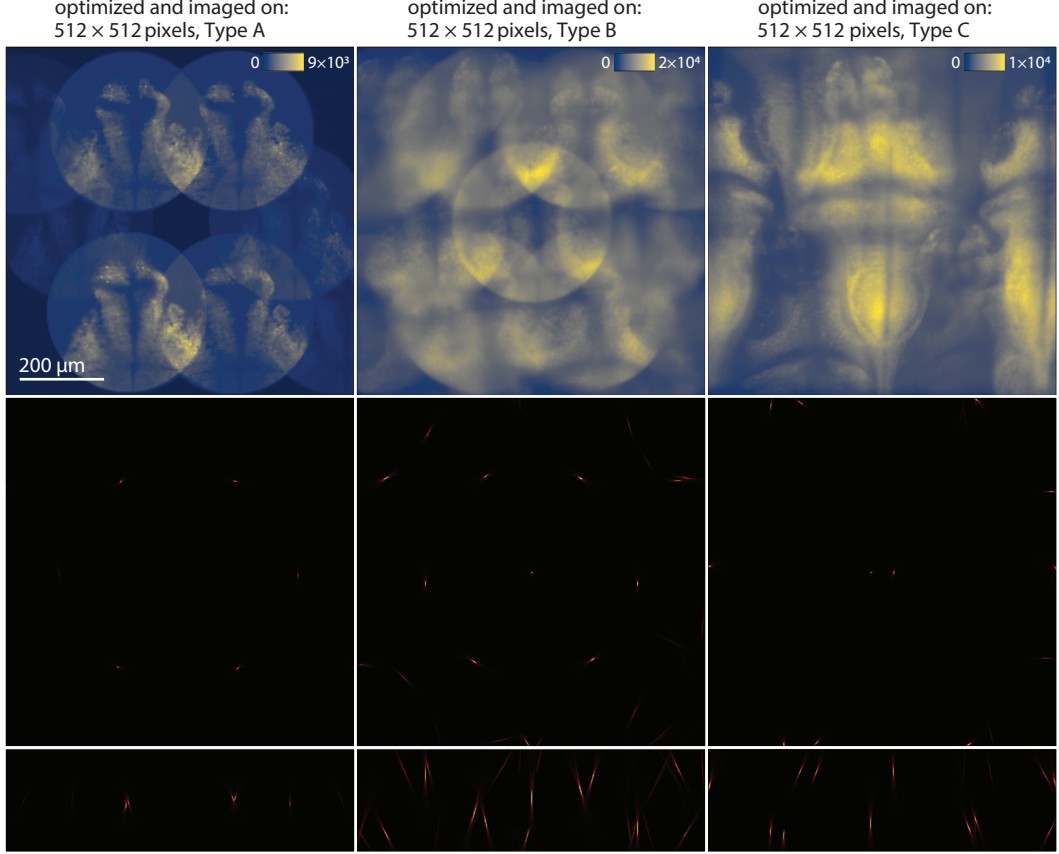

Figure 9: Optimizing optical encoders (PSFs) for different ROIs result in PSFs tailored to each ROI. Note that optical encoder optimized for Type A (left) has pencils with a span in $z$ that matches Type A. Optical encoder optimized for Type B (middle) has pencils that span the entire $z$ depth. Optical encoder optimized for Type C (right) has pencils spread farther apart to account for the larger ROI. Top row shows simulated camera image of a Type A, B, or C example respectively, middle row shows xy max projection of the optical encoder (PSF), and bottom row shows xz max projection of the optical encoder.

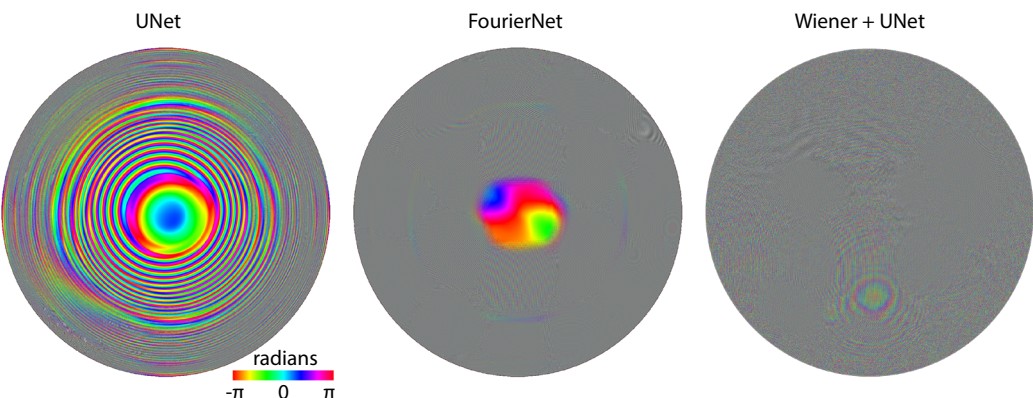

Figure 10: Phase masks (microscope parameters $\boldsymbol{\phi}$) for FourierNet versus UNet. Note that while both phase masks are high-enough frequency to make viewing all pixels difficult after resizing for display and cause their appearance to be gray, the UNet phase mask is much smoother (lower frequency) than the FourierNet phase mask, resulting in a more local optical encoder.

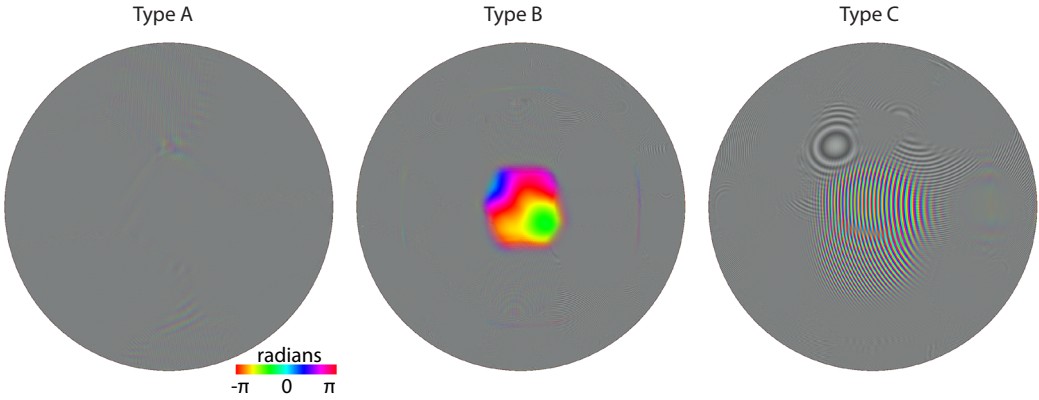

Figure 11: Phase masks (microscope parameters $\boldsymbol{\phi}$) for microscopes optimized for Type A, B, C ROIs, respectively. Note that all phase masks are higher frequency than can be displayed after resizing for this figure, which results in the gray appearance of these phase masks.

only exception to these augmentations is Type C, where we set all the volumes to the same in-plane vertical orientation (while still applying rotation augmentations in pitch and roll).

Table 6: Specifications of all zebrafish datasets Type A, B, C for reconstruction

| Dataset | Camera (px) | Height (planes) | Span (z, y, x) (μm) | Aperture Diameter (μm) |
|---------|-------------|-----------------|---------------------|------------------------|
| Type A | $512 \times 512$ | 12 | (25, 832, 832) | 386 |
| Type B | $512 \times 512$ | 128 | (250, 832, 832) | 386 |
| Type C | $512 \times 512$ | 128 | (250, 832, 832) | - |

**Parallelizing imaging and reconstruction**

We show our parallelization strategy for both imaging and reconstruction in Figure 12 as well as in the following algorithms. Because this simulation can become too expensive in memory to fit on a single device, we generally perform the simulation, reconstruction, and loss calculation in parallel for both training modes. Therefore, any variable that has a $_s$ subscript refers to a list of chunks of that variable that will be run on each device. A $^j$ superscript indicates a particular chunk for GPU $j$. For example $z_s$ is a list of plane indices to be imaged/reconstructed, and $z_s^j$ is the $j^{\text{th}}$ chunk of plane indices that will be imaged/reconstructed on GPU $j$. We denote `parallel` for any operations that are performed in parallel and `scatter` for splitting data into chunks and spreading across multiple GPUs. Imaging can be cleanly parallelized: chunks of an optical encoder (PSF) and sample can be partially imaged on multiple GPUs independently because the convolution occurs per plane, then finally all partial images can be summed together onto a single GPU. The reconstructions can similarly take the final image and reconstruct partial chunks (as well as calculate losses on partial chunks) of the volume independently per device. We implicitly gather data to the same GPU when computing sums ($\sum$) or means ($\mathbb{E}$). The functions `parallel image` and `compute PSF` follow the definitions above in equations 13 and 11. In the algorithms shown here, `parallel image` applies the same convolution described above in equation 13.

**Sparse gradients and downsampling** We additionally support training and reconstructing only some of the planes for imaging and potentially a different partial group of planes during reconstruction, as a way to sparsely compute gradients for optimization of $\theta_m$ and save memory. The planes not imaged with gradients can still contribute to the image (without their gradients being tracked) in order to make the problem more difficult for the reconstruction network. Over multiple iterations, this can become equivalent to the more expensive densely computed gradient method, essentially trading training time for memory. An additional memory saving measure not written in the algorithms is to compute the optical encoder (PSF) at a high resolution, then downsample the optical encoder using a

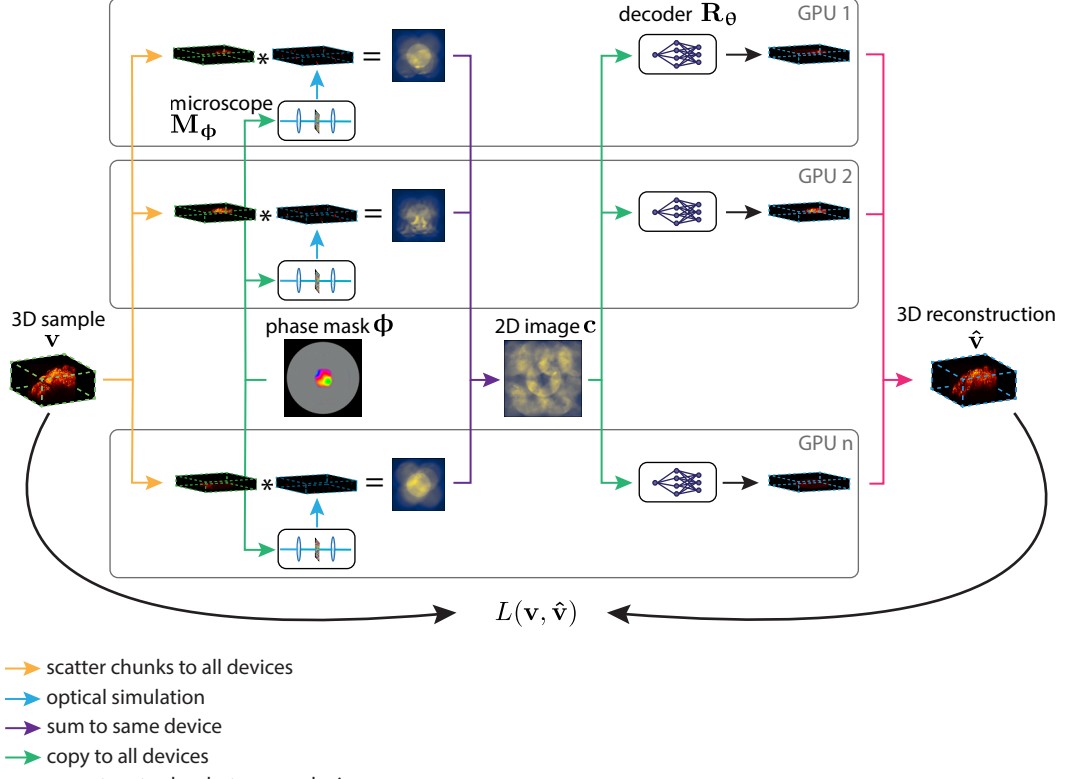

scatter chunks to all devices
optical simulation
sum to same device
copy to all devices
concatenate chunks to same device

Figure 12: Overview of our parallelization strategy, visualizing Algorithm 1. Colored arrows describe parallelization communication, reduction, or computation. The volume is split into chunks, which are scattered to all compute devices (GPUs). Gray boxes demarcate each independent chunk being processed in parallel on its own device. Optical encoder (PSF) computation is simulated for each chunk in parallel, followed by parallel simulation of the imaging as a convolution of the optical encoder and sample chunk. Then, the partial images are summed onto the same device (GPU 1). After simulating shot noise on this single device, the final image is copied to all devices and each chunk is reconstructed on each device in parallel. At this point the loss can be partially computed in parallel for each chunk and then summed for the loss between the full volumes, or the chunked reconstruction can be concatenated to the same device. We show the loss computation at once rather than in parallel for simplification of visualization only. Contrasts for volumes and optical encoders in chunks have been artificially boosted for visibility. Zoom in for best viewing.

2D sum pool to preserve total energy in order to reduce memory usage when performing the imaging and reconstruction. We denote **with no gradient tracking** to show an operation without gradients.

### A.5 Comparing number of optical encoder voxels in simulation to previous works

We compare our optical encoders (PSFs), which are simulated at a maximum size of $64 \times 3840 \times 3840$ voxels in z, y, and x (prior to downsampling for the simulation of imaging) respectively to those of deep learning optical encoder optimizations in localization microscopy and depth from defocus [6, 3]. We simulate at such high voxel counts in order to allow the phase mask (at a size of $3840 \times 3840$ voxels) to produce high frequencies, which are required for producing features in the optical encoder near the edges of the field of view. For localization microscopy, the optical encoders are $51 \times 329 \times 329$ voxels, which means our optical encoder has approximately $170\times$ more voxels [6]. For a state of the art depth from defocus implementation, the optical encoder is simulated with rotational symmetry, which means the actual simulation can occur in only one dimension per depth plane [3]. Thus, the depth from defocus optical encoder is simulated with $16 \times 8000$ voxels in z and x, respectively [3]. The single dimension along x is then rotated to produce the full optical encoder at each of the 16 depth planes [3]. This means our optical encoder has approximately $7372\times$ more unique voxels.

**Algorithm 1:** Parallel optical encoder (PSF) engineering by joint training of reconstruction network and phase mask. Microscope $\mathbf{M}_{\boldsymbol{\phi}}$ parameters are $\boldsymbol{\phi}$, reconstruction network $\mathbf{R}_{\boldsymbol{\theta}}$ parameters are $\boldsymbol{\theta}$, dataset is $\mathbf{D}$, learning rates for $\boldsymbol{\phi}$ and $\boldsymbol{\theta}$ are $\alpha_{\boldsymbol{\phi}}$ and $\alpha_{\boldsymbol{\theta}}$ respectively, plane indices to image and reconstruct from $z_s$, and weight for $L_{\text{NMSE}}$ is $\beta$.

**Input :** $\mathbf{M}_{\boldsymbol{\phi}}, \boldsymbol{\phi}, \alpha_{\boldsymbol{\phi}}, \mathbf{R}_{\boldsymbol{\theta}}, \boldsymbol{\theta}, \alpha_{\boldsymbol{\theta}}, \mathbf{D}, z_s, \beta$

1 **for** $\mathbf{v} \in \mathbf{D}$ **do**
    *// select plane indices to be imaged with and without gradients*
2     $z_{s,\text{gradient}}, z_{s,\text{no gradient}} \leftarrow$ **select planes(** $z_s$ **)**
    *// move sample planes to be imaged with gradients to multiple GPUs*
3     $\mathbf{v}_{s,\text{gradient}} \leftarrow$ **scatter(** $\mathbf{v}, z_{s,\text{gradient}}$ **)**
    *// move sample planes to be imaged without gradients to multiple GPUs*
4     $\mathbf{v}_{s,\text{no gradient}} \leftarrow$ **scatter(** $\mathbf{v}, z_{s,\text{no gradient}}$ **)**
    *// compute PSF with gradients on multiple GPUs*
5     $\mathbf{s}_{s,\text{gradient}} \leftarrow$ **parallel(compute PSF(** $\mathbf{M}_{\boldsymbol{\phi}}, z_s^j$ **) for** $z_s^j$ **in** $z_{s,\text{gradient}}$ **)**
    *// compute partial image with gradients on multiple GPUs*
6     $\mathbf{c}_{\text{gradient}} \leftarrow$ **parallel image(** $\mathbf{s}_{s,\text{gradient}}, \mathbf{v}_{s,\text{gradient}}$ **)**
    *// compute PSF without gradients on multiple GPUs*
7     **with no gradient tracking**
8         $\mathbf{s}_{s,\text{no gradient}} \leftarrow$ **parallel(compute PSF(** $\mathbf{M}_{\boldsymbol{\phi}}, z_s^j$ **) for** $z_s^j$ **in** $z_{s,\text{no gradient}}$ **)**
9     **end**
    *// compute partial image without gradients on multiple GPUs*
10    **with no gradient tracking**
11        $\mathbf{c}_{\text{no gradient}} \leftarrow$ **parallel image(** $\mathbf{s}_{s,\text{no gradient}}, \mathbf{v}_{s,\text{no gradient}}$ **)**
12    **end**
    *// compute full image by summing partial images onto one GPU*
13    $\mathbf{c} \leftarrow \sum [\mathbf{c}_{\text{gradient}}, \mathbf{c}_{\text{no gradient}}]$
    *// select plane indices to be reconstructed*
14    $z_{s,\text{reconstruct}} \leftarrow$ **select planes(** $z_s$ **)**
    *// move sample planes that will be reconstructed to multiple GPUs*
15    $\mathbf{v}_{s,\text{reconstruct}} \leftarrow$ **scatter(** $\mathbf{v}, z_{s,\text{reconstruct}}$ **)**
    *// compute mean of high passed sample for loss normalization*
16    $\mu_{H(\mathbf{v})} \leftarrow \mathbb{E}(H(\mathbf{v}_{s,\text{reconstruct}})^2)$
    *// compute mean of sample for loss normalization*
17    $\mu_{\mathbf{v}} \leftarrow \mathbb{E}(\mathbf{v}_{s,\text{reconstruct}}^2)$
    *// move reconstruction networks to multiple GPUs*
18    $\mathbf{R}_{\boldsymbol{\theta},s} \leftarrow$ **scatter(** $\mathbf{R}_{\boldsymbol{\theta}}$ **)**
    *// compute reconstruction and loss on multiple GPUs*
19    $L \leftarrow$ **parallel reconstruct/loss(** $\mathbf{c}, \mathbf{v}_{s,\text{reconstruct}}, \mathbf{R}_{\boldsymbol{\theta},s}, \mu_{H(\mathbf{v})}, \mu_{\mathbf{v}}, \beta$ **)**
    *// compute gradients for all parameters*
20    $g_{\boldsymbol{\theta}} \leftarrow \nabla_{\boldsymbol{\theta}} L$
21    $g_{\boldsymbol{\phi}} \leftarrow \nabla_{\boldsymbol{\phi}} L$
    *// update all parameters*
22    $\boldsymbol{\theta} \leftarrow$ **Adam(** $\alpha_{\boldsymbol{\theta}}, \boldsymbol{\theta}, g_{\boldsymbol{\theta}}$ **)**
23    $\boldsymbol{\phi} \leftarrow$ **Adam(** $\alpha_{\boldsymbol{\phi}}, \boldsymbol{\phi}, g_{\boldsymbol{\phi}}$ **)**
24 **end**

### A.6 Implementation details

**Fourier convolution details** Our Fourier convolution uses complex number weights, implemented as two channels of real numbers. Furthermore, in order to prevent the convolution from wrapping around the edges, we have to pad the input to double the size. The size of the weight must match the size of this padded input. This means that the number of parameters for our Fourier convolution implementation is $8\times$ the number of parameters required for a global kernel in a spatial convolution (though the Fourier convolution is significantly faster). We do this to save an extra padding and

**Algorithm 2:** Parallel training a reconstruction network given a pre-trained phase mask. Microscope $\mathbf{M_\phi}$ parameters are $\phi$ (phase mask), reconstruction network $\mathbf{R_\theta}$ parameters are $\theta$, dataset is $\mathbf{D}$, learning rates for $\phi$ and $\theta$ are $\alpha_\phi$ and $\alpha_\theta$ respectively, plane indices to image and reconstruct from are $z_s$, and weight for $L_{\mathrm{NMSE}}$ is $\beta$.

---

**Input :** $\mathbf{M_\phi}, \phi, \alpha_\phi, \mathbf{R_\theta}, \theta, \alpha_\theta, \mathbf{D}, z_s, \beta$
    *// compute PSF without gradients on multiple GPUs*
1 **with no gradient tracking**
2    $\mathbf{s}_{\mathrm{no\ gradient}} \leftarrow$ **parallel(compute PSF(**$\mathbf{M_\phi}, z_s^j$**) for** $z_s^j$ **in** $z_s$**)**
3 **end**
4 **for** $\mathbf{v} \in \mathbf{D}$ **do**
      *// select plane indices to be imaged without gradients*
5    $z_{s,\mathrm{no\ gradient}} \leftarrow$ **select planes(**$z_s$**)**
      *// move sample planes to be imaged without gradients to multiple*
         *GPUs*
6    $\mathbf{v}_{s,\mathrm{no\ gradient}} \leftarrow$ **scatter(**$\mathbf{v}, z_{s,\mathrm{no\ gradient}}$**)**
      *// move necessary PSF planes to multiple GPUs*
7    $\mathbf{s}_{s,\mathrm{no\ gradient}} \leftarrow$ **scatter(**$\mathbf{s}_{\mathrm{no\ gradient}}, z_{s,\mathrm{no\ gradient}}$**)**
      *// compute image without gradients on multiple GPUs*
8    **with no gradient tracking**
9      $\mathbf{c} \leftarrow$ **parallel image(**$\mathbf{s}_{s,\mathrm{no\ gradient}}, \mathbf{v}_{s,\mathrm{no\ gradient}}$**)**
10   **end**
      *// select plane indices to be reconstructed*
11   $z_{s,\mathrm{reconstruct}} \leftarrow$ **select planes(**$z_s$**)**
      *// move sample planes that will be reconstructed to multiple GPUs*
12   $\mathbf{v}_{s,\mathrm{reconstruct}} \leftarrow$ **scatter(**$\mathbf{v}, z_{s,\mathrm{reconstruct}}$**)**
      *// compute mean of high passed sample for loss normalization*
13   $\mu_{H(\mathbf{v})} \leftarrow \mathbb{E}[H(\mathbf{v}_{s,\mathrm{reconstruct}})^2]$
      *// compute mean of sample for loss normalization*
14   $\mu_{\mathbf{v}} \leftarrow \mathbb{E}[\mathbf{v}_{s,\mathrm{reconstruct}}^2]$
      *// move reconstruction networks to multiple GPUs*
15   $\mathbf{R}_{\theta,s} \leftarrow$ **scatter(**$\mathbf{R_\theta}$**)**
      *// compute reconstruction and loss on multiple GPUs*
16   $L \leftarrow$ **parallel reconstruct/loss(**$\mathbf{c}, \mathbf{v}_{s,\mathrm{reconstruct}}, \mathbf{R}_{\theta,s}, \mu_{H(\mathbf{v})}, \mu_{\mathbf{v}}, \beta$**)**
      *// compute gradients for reconstruction networks only*
17   $g_\theta \leftarrow \nabla_\theta L$
      *// update reconstruction network parameters only*
18   $\theta \leftarrow$ **Adam(**$\alpha_\theta, \theta, g_\theta$**)**
19 **end**

---

**Algorithm 3:** Parallel imaging. Optical encoder (PSF) planes on multiple GPUs are $\mathbf{s}_s$, sample planes on multiple GPUs to be imaged are $\mathbf{v}_s$.

---

**Input**  : $\mathbf{s}_s, \mathbf{v}_s$
**Output** : $\mathbf{c}$
*// compute images in parallel on multiple GPUs, then sum to single GPU*
1 $\mathbf{c} \leftarrow \sum$[**parallel(convolve(**$\mathbf{s}_\mathbf{s}^\mathbf{j}, \mathbf{v}_\mathbf{s}^\mathbf{j}$**) for** $(\mathbf{s}_s^j, \mathbf{v}_\mathbf{s}^\mathbf{j})$ **in** $(\mathbf{s}_s, \mathbf{v}_s)$**)**]
2 **return** $\mathbf{c}$

---

Fourier operation, trading memory for speed. Because the simulation of imaging requires more memory than the reconstruction network, we found this to be an acceptable tradeoff.

**Common network details** All convolutions (including Fourier convolutions) use "same" padding. For FourierUNets and vanilla UNets, downsampling and upsampling is performed only in the $x$ and $y$ dimensions (we do not downsample or upsample in $z$ because there could potentially not be enough planes to do so). We train all networks using the ADAM optimizer with all default PyTorch parameters

**Algorithm 4:** Parallel reconstruction/loss calculation. Camera image is $\mathbf{c}$, sample planes on multiple GPUs are $\mathbf{v}_s$, reconstruction networks on multiple GPUs are $\mathbf{R}_{\boldsymbol{\theta},s}$, mean for $L_{\text{HNMSE}}$ normalization is $\mu_{H(\mathbf{v})}$, mean for $L_{\text{NMSE}}$ normalization is $\mu_{\mathbf{v}}$, and weight for $L_{\text{NMSE}}$ is $\beta$.

---

**Input** : $\mathbf{c}, \mathbf{v}_s, \mathbf{R}_{\boldsymbol{\theta},s}, \mu_{H(\mathbf{v})}, \mu_{\mathbf{v}}, \beta$
**Output** : L
*// compute reconstruction and loss in parallel on multiple GPUs*
1 $\hat{\mathbf{v}}_s \leftarrow$ **concatenate(parallel(** $\mathbf{R}_s^j(\mathbf{c})$ **for** $\mathbf{R}_s^j$ **in** $\mathbf{R}_{\boldsymbol{\theta},s}$**))**
2 $L_s \leftarrow$ **parallel(** $\frac{\mathbb{E}[(H(\mathbf{v}_s^j) - H(\hat{\mathbf{v}}_s^j))^2]}{\mu_{H(\mathbf{v})}} + \beta \frac{\mathbb{E}[(\mathbf{v}_s^j - \hat{\mathbf{v}}_s^j)^2]}{\mu_{\mathbf{v}}}$ **for** $(\hat{\mathbf{v}}_s^j, \mathbf{v}_s^j)$ **in** $(\hat{\mathbf{v}}_s, \mathbf{v}_s)$**)**
*// compute mean of scattered losses on single GPU*
3 $L \leftarrow \mathbb{E}[L_s]$
4 **return** $L$

---

except the learning rate, which we always set to $10^{-4}$ for the reconstruction network parameters $\boldsymbol{\theta}$ and $10^{-2}$ for the phase mask parameters $\boldsymbol{\phi}$.

**Normalization** We use **input scaling** during both training and inference in order to normalize out differences in the brightness of the image and prevent instabilities in our gradients. This means we divide out the median value of the input (scaled by some factor in order to bring the loss to a reasonable range) and then undo this scaling after the output of the network. This effectively linearizes our reconstruction networks, meaning a scaling of the image sent to the network will exactly scale the output by that value. We also find this is a more effective and simpler alternative to using a BatchNorm on our inputs. We continue to use BatchNorm between our convolution layers within the reconstruction network [41], which is effectively InstanceNorm in our case where batch size is 1 [42].

**Planewise network training logic** When we train optical encoders by optimizing $\boldsymbol{\phi}$, we train separate reconstruction networks per plane. This allows us to flexibly compute sparse gradients across different planes from iteration to iteration, as described in Appendix A.4. In order to do this, we create placeholder networks on any number of GPUs, then copy the parameters stored on CPU for each plane's reconstruction network to a network on the GPU as needed during a forward pass. After calculating an update with the optimizer, we copy the parameter values back to the corresponding parameter on CPU.

**Training times** We optimize our smaller, $256 \times 256$ pixel microscopy experiments on 4 RTX 2080 Ti GPUs when optimizing both $\boldsymbol{\phi}$ and $\boldsymbol{\theta}$ and 8 RTX 2080 Ti GPUs when optimizing only $\boldsymbol{\theta}$, except the Wiener + UNet model which is trained on 8 RTX Quadro 8000 GPUs. For these, we can compare training times for the different network architectures. One training iteration (including microscope simulation, reconstruction, backpropagation, and parameter update) takes $\sim$0.6 seconds for FourierNet2D, $\sim$1.3 seconds for UNet2D, and $\sim$0.8 seconds for Wiener + UNet when optimizing both $\boldsymbol{\phi}$ and $\boldsymbol{\theta}$. One training iteration takes $\sim$0.4 seconds for FourierNet3D, $\sim$0.7 seconds for FourierUNet3D, and $\sim$0.8 seconds for UNet3D when only optimizing $\boldsymbol{\theta}$. Our larger Type A, B, C experiments are always optimized on 8 RTX Quadro 8000 GPUs. More details are found in Tables 7 and 8.

**Memory usage** We show our training GPU memory usage for all kinds of snapshot microscopy experiments training both optical encoders and training reconstruction networks only in Table 9. Because we must synchronize some computations to a single GPU, there will be one GPU with higher memory usage than the rest. Thus, we report both the highest memory usage of a single GPU (the maximum memory usage across GPUs) as well as the memory usage of the remaining single GPUs (the mode memory usage across GPUs).

### A.7 Details for FourierNets outperform state-of-the-art for reconstructing natural images captured by DiffuserCam lensless camera

We performed no augmentations for this set of trainings reconstructing RGB color images of natural scenes from RGB diffused images taken through a DiffuserCam [25]. We modified our FourierNet2D architecture to create the FourierNetRGB architecture and our FourierUNet2D architecture to create the FourierUNetRGB architecture, outlined in Table 11 and Table 12 respectively. Training details are

Table 7: Small experiment training times

| Network | Optimizing | # parameters | # train steps | Train step time (s) | Total time (h) |
|---|---|---|---|---|---|
| FourierNet2D | $\theta, \phi$ | $\sim 4.2 \times 10^7$ | $10^6$ | $\sim 0.8$ | $\sim 222$ |
| FourierNet3D | $\theta$ | $\sim 6.3 \times 10^7$ | $10^6$ | $\sim 0.4$ | $\sim 111$ |
| FourierUNet3D | $\theta$ | $\sim 8.4 \times 10^7$ | $10^6$ | $\sim 0.7$ | $\sim 194$ |
| UNet2D | $\theta, \phi$ | $\sim 4.0 \times 10^7$ | $10^6$ | $\sim 1.3$ | $\sim 361$ |
| Wiener + UNet | $\theta, \phi$ | $\sim 8.0 \times 10^7$ | $5 \times 10^5$ | $\sim 0.8$ | $\sim 111$ |
| UNet3D | $\theta$ | $\sim 1.0 \times 10^8$ | $10^6$ | $\sim 0.8$ | $\sim 222$ |

Table 8: Type A, B, C experiment training times

| Network | Optimizing | # parameters | Type | # train steps | Train step time (s) | Total time (h) |
|---|---|---|---|---|---|---|
| FourierNet2D | $\theta, \phi$ | $\sim 1.7 \times 10^8$ | A | $5.8 \times 10^5$ | $\sim 1.1$ | $\sim 177$ |
| FourierNet3D | $\theta$ (fixed $\phi$ for A) | $\sim 3.4 \times 10^8$ | A | $\sim 2.6 \times 10^5$ | $\sim 1.6$ | $\sim 116$ |
| FourierNet3D | $\theta$ (fixed $\phi$ for A) | $\sim 3.4 \times 10^8$ | B | $\sim 1.3 \times 10^5$ | $\sim 1.6$ | $\sim 58$ |
| FourierNet3D | $\theta$ (fixed $\phi$ for A) | $\sim 3.4 \times 10^8$ | C | $\sim 1.3 \times 10^5$ | $\sim 1.6$ | $\sim 58$ |
| FourierNet2D | $\theta, \phi$ | $\sim 1.7 \times 10^8$ | B | $5.8 \times 10^5$ | $\sim 1.1$ | $\sim 177$ |
| FourierNet3D | $\theta$ (fixed $\phi$ for B) | $\sim 3.4 \times 10^8$ | A | $\sim 1.2 \times 10^5$ | $\sim 1.6$ | $\sim 53$ |
| FourierNet3D | $\theta$ (fixed $\phi$ for B) | $\sim 3.4 \times 10^8$ | B | $10^6$ | $\sim 1.6$ | $\sim 444$ |
| FourierNet3D | $\theta$ (fixed $\phi$ for B) | $\sim 3.4 \times 10^8$ | C | $\sim 5.0 \times 10^5$ | $\sim 1.6$ | $\sim 222$ |
| FourierNet2D | $\theta, \phi$ | $\sim 1.7 \times 10^8$ | C | $5.8 \times 10^5$ | $\sim 1.1$ | $\sim 177$ |
| FourierNet3D | $\theta$ (fixed $\phi$ for C) | $\sim 3.4 \times 10^8$ | A | $\sim 3.4 \times 10^5$ | $\sim 1.6$ | $\sim 151$ |
| FourierNet3D | $\theta$ (fixed $\phi$ for C) | $\sim 3.4 \times 10^8$ | B | $\sim 3.4 \times 10^5$ | $\sim 1.6$ | $\sim 151$ |
| FourierNet3D | $\theta$ (fixed $\phi$ for C) | $\sim 3.4 \times 10^8$ | C | $\sim 3.7 \times 10^5$ | $\sim 1.6$ | $\sim 164$ |

Table 9: GPU memory usage for all snapshot microscopy experiment types

| Network | Optimizing | Type | # GPUs | Max (MB) | Mode (MB) | Total (MB) |
|---|---|---|---|---|---|---|
| FourierNet2D | $\theta, \phi$ | Small | 4 | 4,815 | 4,783 | 19,164 |
| UNet2D | $\theta, \phi$ | Small | 4 | 5,177 | 5,145 | 20,612 |
| Wiener + UNet | $\theta, \phi$ | Small | 4 | 6,465 | 6,253 | 50,860 |
| FourierNet3D | $\theta$ | Small | 8 | 2,647 | 1,617 | 13,966 |
| FourierUNet3D | $\theta$ | Small | 8 | 2,725 | 1,631 | 14,142 |
| UNet3D | $\theta$ | Small | 8 | 2,751 | 1,603 | 13,972 |
| FourierNet2D | $\theta, \phi$ | A, B, C | 8 | 8,513 | 8,267 | 66,382 |
| FourierNet3D | $\theta$ | A, B, C | 8 | 15,537 | 3,865 | 42,592 |

shown in Table 10. Because these reconstructions are of 2D images only and required no microscope simulation, we were able to use a batch size of 4 images per iteration.

Table 10: DLMD experiment training times. Superscripts denote loss function: [1] MSE, [2] MSE+LPIPS.

| Network | Optimizing | # parameters | # train steps | Train step time (s) | Total time (h) |
|---|---|---|---|---|---|
| FourierNetRGB[1] | $\theta$ | $\sim 1.6 \times 10^7$ | $2.2 \times 10^5$ | $\sim 0.43$ | $\sim 26$ |
| FourierNetRGB[2] | $\theta$ | $\sim 1.6 \times 10^7$ | $1.1 \times 10^5$ | $\sim 0.47$ | $\sim 14$ |
| FourierUNetRGB[1] | $\theta$ | $\sim 7.1 \times 10^7$ | $2.5 \times 10^5$ | $\sim 3.3$ | $\sim 229$ |
| Le-ADMM-U[2] [25] | $\theta$ | $\sim 4.0 \times 10^7$ | - | - | - |
| UNet[2] [25] | $\theta$ | $\sim 1.0 \times 10^8$ | - | - | - |

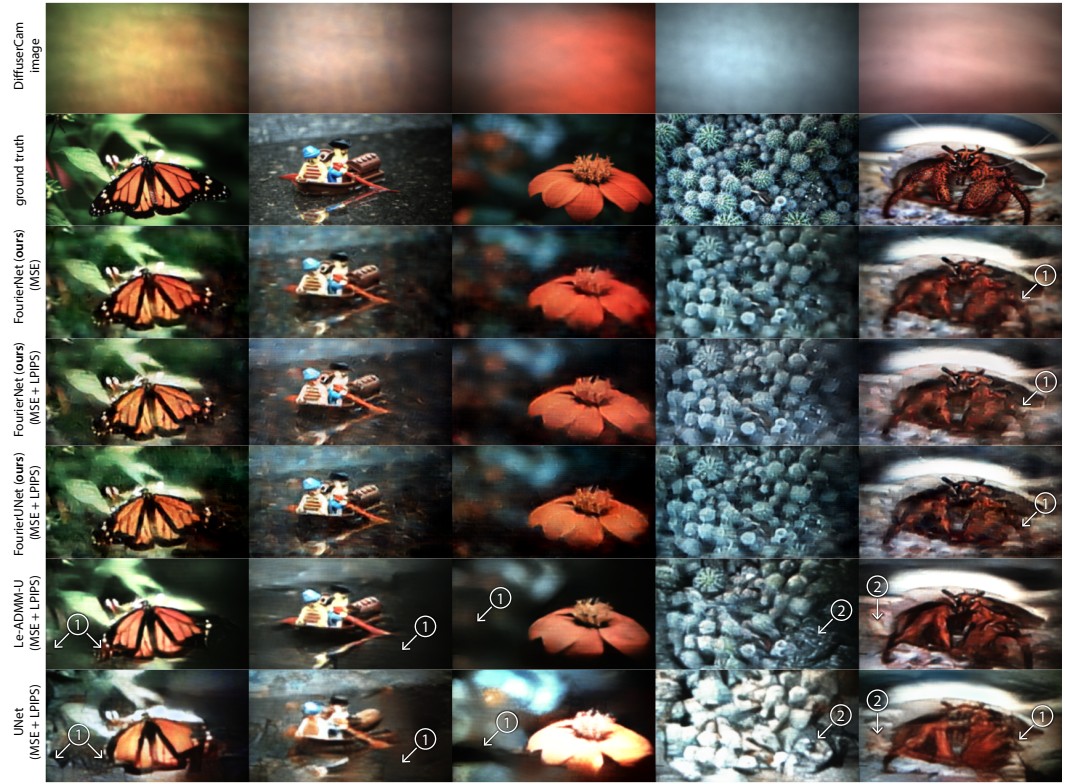

Figure 13: Comparisons of our method (second and third rows) to state-of-the-art learned reconstruction methods on lensless diffused images of natural scenes. Note that FourierNet trained with MSE only shows comparable visual results to training with MSE + LPIPS. Regions labeled ① indicate missing details, either resolution or textures in backgrounds. Regions labeled ② indicate hallucinated textures. Note that the previous state-of-the-art solutions [25] exhibit both issues more often compared to our models.

Table 11: FourierNetRGB detailed architecture

| Layer type | Kernel size | Stride | Notes | Shape (N, C, H, W) |
|---|---|---|---|---|
| FourierConv2D | (270, 480) | (2, 2) | - | (4, 3, 270, 480) |
| LeakyReLU | - | - | slope: -0.01 | (4, 20, 270, 480) |
| BatchNorm2D | - | - | - | (4, 20, 270, 480) |
| Conv2D | (11, 11) | (1, 1) | - | (4, 64, 270, 480) |
| BatchNorm2D | - | - | - | (4, 64, 270, 480) |
| LeakyReLU | - | - | slope: -0.01 | (4, 64, 270, 480) |
| Conv2D | (11, 11) | (1, 1) | - | (4, 64, 270, 480) |
| BatchNorm2D | - | - | - | (4, 64, 270, 480) |
| LeakyReLU | - | - | slope: -0.01 | (4, 64, 270, 480) |
| Conv2D | (11, 11) | (1, 1) | - | (4, 3, 270, 480) |
| ReLU | - | - | - | (4, 3, 270, 480) |

## A.8 Details for FourierNets outperform UNets for engineering non-local optical encoders and 3D snapshot microscopy volume reconstruction

For our experiments in Sections 3.2 and 3.3, we use 40 planes at 5 $\mu$m resolution in z and therefore 40 reconstruction networks to train PSFs, except the Wiener + UNet model which is trained in a single stage. When training reconstruction networks only to produce the higher quality reconstructions, we use 96 planes at 1 $\mu$m resolution in z (chosen so that the planes actually span 200 $\mu$m in z). Following [3], the Wiener + UNet model is only trained in one stage (using knowledge of the current PSF, which the other methods do not receive), and is always trained on 96 planes. We train in both settings

Table 12: FourierUNetRGB detailed architecture

| Scale | Repeat | Layer type | Kernel size | Stride | Notes | Shape (N, C, H, W) |
|---|---|---|---|---|---|---|
| 1 | 1 | Multiscale FourierConv2D + ReLU + BatchNorm2D | (270, 480) | (2, 2) | - | (4, 64, 270, 480) |
| 2 | | | (135, 240) | (2, 2) | | (4, 64, 135, 240) |
| 3 | | | (67, 120) | (2, 2) | | (4, 64, 67, 120) |
| 4 | | | (33, 60) | (2, 2) | | (4, 64, 33, 60) |
| 3 | 1 | Upsample2D | - | - | - | (4, 64, 67, 120) |
| 3 | 2 | Conv2D + ReLU + BatchNorm2D | (11, 11) | (1, 1) | - | (4, 64, 67, 120) |
| 2 | 1 | Upsample2D | - | - | - | (4, 64, 135, 240) |
| 2 | 2 | Conv2D + ReLU + BatchNorm2D | (11, 11) | (1, 1) | - | (4, 64, 135, 240) |
| 1 | 1 | Upsample2D | - | - | - | (4, 64, 270, 480) |
| 1 | 2 | Conv2D + ReLU + BatchNorm2D | (11, 11) | (1, 1) | - | (4, 64, 270, 480) |
| 1 | 1 | Conv2D + ReLU | (1, 1) | (1, 1) | - | (4, 3, 270, 480) |

Table 13: FourierNet2D detailed architecture (1 per plane)

| Layer type | Kernel size | Stride | Notes | Shape (C, D, H, W) |
|---|---|---|---|---|
| InputScaling | - | - | scale: 0.01 | (1, 1, 256, 256) |
| FourierConv2D | (256, 256) | (2, 2) | - | (8, 1, 256, 256) |
| LeakyReLU | - | - | slope: -0.01 | (8, 1, 256, 256) |
| BatchNorm2D | - | - | - | (8, 1, 256, 256) |
| Conv2D | (11, 11) | (1, 1) | - | (1, 1, 256, 256) |
| ReLU | - | - | - | (1, 1, 256, 256) |
| InputRescaling | - | - | scale: 0.01 | (1, 1, 256, 256) |

Table 14: FourierNet3D detailed architecture (8 GPUs)

| Layer type | Kernel size | Stride | Notes | Shape (C, D, H, W) |
|---|---|---|---|---|
| InputScaling | - | - | scale: 0.01 | (1, 1, 256, 256) |
| FourierConv2D | (256, 256) | (2, 2) | - | (60, 1, 256, 256) |
| LeakyReLU | - | - | slope: -0.01 | (60, 1, 256, 256) |
| BatchNorm2D | - | - | - | (60, 1, 256, 256) |
| Reshape2D3D | - | - | - | (5, 12, 256, 256) |
| Conv3D | (11, 7, 7) | (1, 1, 1) | - | (5, 12, 256, 256) |
| LeakyReLU | - | - | slope: -0.01 | (5, 12, 256, 256) |
| BatchNorm3D | - | - | - | (5, 12, 256, 256) |
| Conv3D | (11, 7, 7) | (1, 1, 1) | - | (1, 12, 256, 256) |
| ReLU | - | - | - | (1, 12, 256, 256) |
| InputRescaling | - | - | scale: 0.01 | (1, 12, 256, 256) |

without any sparse planewise gradients, meaning we image and reconstruct all 40 or all 96 planes, respectively. We show details of all datasets used for training reconstructions in Table 6.

We show the details of our FourierNet2D architecture for training PSFs in Table 13 and our Fourier-Net3D architecture for training reconstruction networks in Table 14. We also show details for training times for both training PSFs and for training more powerful reconstruction networks in Table 7. We

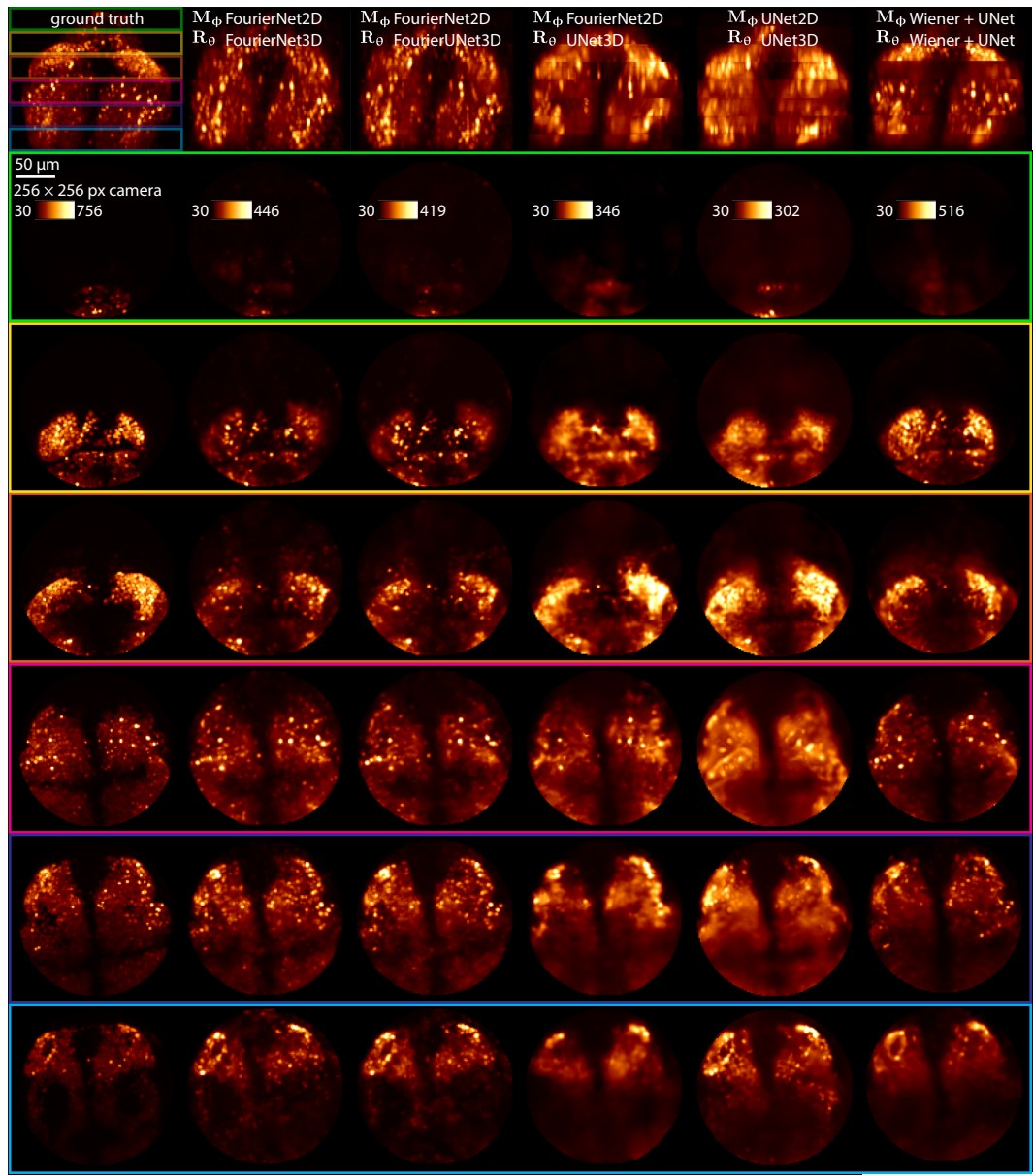

Figure 14: Slab views of a small example volume reconstruction, showing our methods (Fourier-Net/FourierUNet) do the best job of reconstructing throughout the volume. Note that the UNet reconstructions are blurry across all slabs, with few exceptions, while the Wiener + UNet reconstruction is not blurry in some slabs, but not consistent across all slabs. Colored boxes show which sample planes a particular slab comes from, corresponding to boxes in xz projection view at top. Annotation $\mathbf{M_\phi}$ shows which network architecture was used for phase mask optimization; annotation $\mathbf{R_\theta}$ shows which architecture was used for reconstruction.

trained all networks for small $256 \times 256$ pixel experiments for the same number of iterations (more than necessary for PSFs to meaningfully converge)[2].

The architecture of FourierUNet3D is 4 scales, with a cropping factor of 2 per scale in the encoding path and an upsampling factor of 2 in the decoding path. For each scale, we perform a Fourier convolution in the encoding path producing 480 feature maps, which are concatenated with the

---

[2]Training times are approximate, and actual total time was longer due to checkpointing/snapshotting/validation of data and/or differences in load on the clusters being used.

Table 15: FourierUNet3D detailed architecture (8 GPUs)

| Scale | Repeat | Layer type | Kernel size | Stride | Notes | Shape (C, D, H, W) |
|---|---|---|---|---|---|---|
| 1 | 1 | InputScaling | - | - | scale: 0.01 | (1, 1, 256, 256) |
| 1 | 1 | Multiscale FourierConv2D + ReLU + BatchNorm2D | (256, 256) | (2, 2) | - | (60, 1, 256, 256) |
| 2 | | | (128, 128) | (2, 2) | | (60, 1, 128, 128) |
| 3 | | | (64, 64) | (2, 2) | | (60, 1, 64, 64) |
| 4 | | | (32, 32) | (2, 2) | | (60, 1, 32, 32) |
| 4 | 1 | Reshape2D3D | - | - | - | (5, 12, 32, 32) |
| 3 | 1 | Upsample2D | - | - | - | (5, 12, 64, 64) |
| 3 | 2 | Conv3D + ReLU + BatchNorm3D | (11, 7, 7) | (1, 1, 1) | - | (5, 12, 64, 64) |
| 2 | 1 | Upsample2D | - | - | - | (5, 12, 128, 128) |
| 2 | 2 | Conv3D + ReLU + BatchNorm3D | (11, 7, 7) | (1, 1, 1) | - | (5, 12, 128, 128) |
| 1 | 1 | Upsample2D | - | - | - | (5, 12, 256, 256) |
| 1 | 2 | Conv3D + ReLU + BatchNorm3D | (11, 7, 7) | (1, 1, 1) | - | (5, 12, 256, 256) |
| 1 | 1 | Conv3D + ReLU | (1, 1, 1) | (1, 1, 1) | - | (1, 12, 256, 256) |
| 1 | 1 | InputRescaling | - | - | scale: 0.01 | (1, 12, 256, 256) |

incoming feature maps of the decoding convolutions at the corresponding scale (just as in a normal UNet). In the decoding path, we use 3D convolutions with kernel size (3, 5, 5), producing 12 3D feature maps each. There are two such convolutions per scale. Note that this requires we reshape the 2D feature maps from the Fourier convolutions to 3D. This is followed by a 1x1 convolution producing the 3D reconstruction output. We show a diagram of this architecture in Figure 1C, and details of this architecture in Table 15.

For our UNet2D, each encoding convolution produced 24 feature maps (except the first scale, for which the first convolution produced 12 feature maps and the second convolution produced 24 feature maps). Each decoding convolution produced 24 feature maps, but took an input of 48 feature maps where 24 feature maps were concatenated from the corresponding encoding convolution at that scale. At the end of the UNet2D, a (1, 1) convolution reduced the 24 final feature maps to 1 feature map. This single feature map is interpreted as the final output of the network, i.e. the reconstructed plane. UNet2D requires many more feature maps per plane and more layers than FourierNet, because these are necessary in order for the network to be able to integrate information from a larger field of view. The effective field of view is $4,539 \times 4,539$ pixels. We show the details of our UNet2D architecture in Table 16.

The Wiener + UNet model first performs a Wiener deconvolution on the image using the PSF computed from the phase mask, then applies a UNet with a different 3D architecture, following that of [27] which was designed to refine a Wiener filter. We show the details of this Wiener + UNet architecture in Table 18.

The architecture of the vanilla UNet3D is also 4 scales, with a max pooling factor of 2 per scale in the encoding path and an upsampling factor of 2 in the decoding path. Each scale of the encoding path produces 480 2D feature maps. These are concatenated to the incoming feature maps of the decoding convolutions at the corresponding scale, again with a reshape from 2D to 3D. Each scale of the decoding path produces 48 3D feature maps. Again, this is followed by a 1x1 convolution producing the 3D reconstruction output. All convolutions are in 3D with a kernel size of (5, 7, 7), with the $z$ dimension being ignored for the encoding path because the input is 2D. UNet3D has a

Table 16: UNet2D detailed architecture (1 per plane)

| Scale | Repeat | Layer type | Kernel size | Stride | Notes | Shape (C, D, H, W) |
|---|---|---|---|---|---|---|
| 1 | 1 | InputScaling | - | - | scale: 0.01 | (1, 1, 256, 256) |
| 1 | 1 | Conv2D
+ ReLU
+ BatchNorm2D | (7, 7) | (1, 1) | - | (12, 1, 256, 256) |
| 1 | 1 | Conv2D
+ ReLU
+ BatchNorm2D | (7, 7) | (1, 1) | - | (24, 1, 256, 256) |
| 2 | 1 | MaxPool2D | (2, 2) | (2, 2) | - | (24, 1, 128, 128) |
| 2 | 2 | Conv2D
+ ReLU
+ BatchNorm2D | (7, 7) | (1, 1) | - | (24, 1, 128, 128) |
| $n$ | 1 | MaxPool2D | (2, 2) | (2, 2) | - | $(24, 1, \frac{256}{2^{n-1}}, \frac{256}{2^{n-1}})$ |
| $n$ | 2 | Conv2D
+ ReLU
+ BatchNorm2D | (7, 7) | (1, 1) | - | $(24, 1, \frac{256}{2^{n-1}}, \frac{256}{2^{n-1}})$ |
| 8 | 1 | MaxPool2D | (2, 2) | (2, 2) | - | (24, 1, 2, 2) |
| 8 | 2 | Conv2D
+ ReLU
+ BatchNorm2D | (7, 7) | (1, 1) | - | (24, 1, 2, 2) |
| 7 | 1 | Upsample2D | - | - | - | (24, 1, 4, 4) |
| 7 | 2 | Conv2D
+ ReLU
+ BatchNorm2D | (7, 7) | (1, 1) | - | (24, 1, 4, 4) |
| $n$ | 1 | Upsample2D | - | - | - | $(24, 1, \frac{256}{2^{n-1}}, \frac{256}{2^{n-1}})$ |
| $n$ | 2 | Conv2D
+ ReLU
+ BatchNorm2D | (7, 7) | (1, 1) | - | $(24, 1, \frac{256}{2^{n-1}}, \frac{256}{2^{n-1}})$ |
| 1 | 1 | Upsample2D | - | - | - | (24, 1, 256, 256) |
| 1 | 2 | Conv2D
+ ReLU
+ BatchNorm2D | (7, 7) | (1, 1) | - | (24, 1, 256, 256) |
| 1 | 1 | Conv2D
+ ReLU | (1, 1) | (1, 1) | - | (1, 1, 256, 256) |
| 1 | 1 | InputRescaling | - | - | scale: 0.01 | (1, 1, 256, 256) |

global receptive field of $279 \times 279$ pixels. We show the details of our UNet3D architecture in Table 17.

## A.9 Details for engineered optical encoding depends on region of interest

For our experiments in Section 3.4, we use 64 planes at 1 $\mu$m resolution in z and therefore 64 reconstruction networks to train PSFs. When training reconstruction networks only to produce the higher quality reconstructions, we use 128 planes at 1 $\mu$m resolution in z (chosen so that the planes actually span 250 $\mu$m in z). We train in the reconstruction only setting without any sparse planewise gradients, meaning we image and reconstruct all 128 planes. However, when training a PSF we image and reconstruct 40 planes at a time with gradient per iteration (spread across 8 GPUs). These 40 planes are chosen randomly at every iteration from the 64 total possible planes, making potentially separate draws of planes for imaging and reconstruction. We show details of all datasets used for training reconstructions in Table 6.

We show the details of our FourierNet2D architecture for training PSFs at the larger field of view in Type A, B, C in Table 19 and our FourierNet3D architecture for training reconstruction networks at the larger field of view in Type A, B, C in Table 20. There are no other networks used for these larger field of view experiments. We also show details for training times for both training PSFs and for

Table 17: UNet3D detailed architecture (8 GPUs)

| Scale | Repeat | Layer type | Kernel size | Stride | Notes | Shape (C, D, H, W) |
|---|---|---|---|---|---|---|
| 1 | 1 | InputScaling | - | - | scale: 0.01 | (1, 1, 256, 256) |
| 1 | 1 | Conv2D
+ ReLU
+ BatchNorm2D | (7, 7) | (1, 1) | - | (30, 1, 256, 256) |
| 1 | 1 | Conv2D
+ ReLU
+ BatchNorm2D | (7, 7) | (1, 1) | - | (60, 1, 256, 256) |
| 2 | 1 | MaxPool2D | (2, 2) | (2, 2) | - | (60, 1, 128, 128) |
| 2 | 2 | Conv2D
+ ReLU
+ BatchNorm2D | (7, 7) | (1, 1) | - | (60, 1, 128, 128) |
| 3 | 1 | MaxPool2D | (2, 2) | (2, 2) | - | (60, 1, 64, 64) |
| 3 | 2 | Conv2D
+ ReLU
+ BatchNorm2D | (7, 7) | (1, 1) | - | (60, 1, 64, 64) |
| 4 | 1 | MaxPool2D | (2, 2) | (2, 2) | - | (60, 1, 32, 32) |
| 4 | 2 | Conv2D
+ ReLU
+ BatchNorm2D | (7, 7) | (1, 1) | - | (60, 1, 32, 32) |
| 4 | 1 | Reshape2D3D | - | - | - | (5, 12, 32, 32) |
| 3 | 1 | Upsample2D | - | - | - | (5, 12, 64, 64) |
| 3 | 2 | Conv3D
+ ReLU
+ BatchNorm3D | (11, 7, 7) | (1, 1, 1) | - | (5, 12, 64, 64) |
| 2 | 1 | Upsample2D | - | - | - | (5, 12, 128, 128) |
| 2 | 2 | Conv3D
+ ReLU
+ BatchNorm3D | (11, 7, 7) | (1, 1, 1) | - | (5, 12, 128, 128) |
| 1 | 1 | Upsample2D | - | - | - | (5, 12, 256, 256) |
| 1 | 2 | Conv3D
+ ReLU
+ BatchNorm3D | (11, 7, 7) | (1, 1, 1) | - | (5, 12, 256, 256) |
| 1 | 1 | Conv3D
+ ReLU | (1, 1, 1) | (1, 1, 1) | - | (1, 12, 256, 256) |
| 1 | 1 | InputRescaling | - | - | scale: 0.01 | (1, 12, 256, 256) |

training more powerful reconstruction networks in Table 8. All PSFs in these networks were trained for the same number of iterations. However, reconstruction networks for some of these experiments were only trained for as long as necessary to converge (with some exceptions where we attempted longer training to check for performance gains with long training periods). Generally, we observed that performance for such reconstruction networks does not meaningfully change with many more iterations of training[3].

## A.10  Details for engineered optical encoders implemented on a programmable microscope

The experimental data presented in this manuscript was acquired with a prototype programmable microscope. Light was collected by a 16X, 0.8 NA microscope objective (N16XLWD-PF, Nikon) and relayed onto an image plane by a 200 mm tube lens (TL200CLS2, Thorlabs). A polarizing beam splitter (PBS251, Thorlabs) transmitted horizontally polarized light. A set of tube lenses relayed the back pupil of the objective lens onto a spatial light modulator (P1920-532, Meadowlark), which was used to modulate the phase of the collected fluorescence with the optimized phase mask. The modulated light was imaged onto an sCMOS camera (Orca-Flash4.0 C11440, Hamamatsu). The total

---

[3]Training times are approximate, and actual total time was longer due to checkpointing/snapshotting/validation of data and/or differences in load on the clusters being used.

Table 18: Wiener + UNet detailed architecture (8 GPUs)

| Scale | Repeat | Layer type | Kernel size | Stride | Notes | Shape (C, D, H, W) |
|---|---|---|---|---|---|---|
| 1 | 1 | InputScaling | - | - | scale: 0.01 | (1, 1, 256, 256) |
| 1 | 1 | WienerFilter | - | - | - | (12, 1, 256, 256) |
| 1 | 1 | Reshape2D3D | - | - | - | (1, 12, 256, 256) |
| 1 | 1 | Conv3D + LeakyReLU | (3, 3, 3) | (1, 1) | - | (32, 12, 256, 256) |
| 1 | 1 | Conv3D + LeakyReLU | (3, 3, 3) | (1, 1) | - | (32, 12, 256, 256) |
| 2 | 1 | Conv3D | (2, 2, 2) | (2, 2, 2) | - | (32, 6, 128, 128) |
| 2 | 2 | Conv3D + LeakyReLU | (3, 3, 3) | (1, 1) | - | (64, 6, 128, 128) |
| 3 | 1 | Conv3D | (2, 2, 2) | (2, 2, 2) | - | (64, 3, 64, 64) |
| 3 | 2 | Conv3D + LeakyReLU | (3, 3, 3) | (1, 1) | - | (128, 3, 64, 64) |
| 4 | 1 | Conv3D | (2, 2, 2) | (2, 2, 2) | - | (128, 1, 32, 32) |
| 4 | 2 | Conv3D + LeakyReLU | (3, 3, 3) | (1, 1) | - | (256, 1, 32, 32) |
| 5 | 1 | Conv3D | (2, 2, 2) | (2, 2, 2) | padding (1, 0, 0) | (256, 1, 16, 16) |
| 5 | 2 | Conv3D + LeakyReLU | (3, 3, 3) | (1, 1) | - | (256, 1, 16, 16) |
| 4 | 1 | ConvTranspose3D | (1, 2, 2) | (2, 2, 2) | - | (256, 1, 32, 32) |
| 4 | 2 | Conv3D + LeakyReLU | (3, 3, 3) | (1, 1) | - | (128, 1, 32, 32) |
| 3 | 1 | ConvTranspose3D | (2, 2, 2) | (2, 2, 2) | output padding (1, 0, 0) | (64, 3, 64, 64) |
| 3 | 2 | Conv3D + LeakyReLU | (3, 3, 3) | (1, 1) | - | (64, 3, 64, 64) |
| 2 | 1 | ConvTranspose3D | (2, 2, 2) | (2, 2, 2) | - | (64, 6, 128, 128) |
| 2 | 2 | Conv3D + LeakyReLU | (3, 3, 3) | (1, 1) | - | (32, 6, 128, 128) |
| 1 | 1 | ConvTranspose3D | (2, 2, 2) | (2, 2, 2) | - | (32, 12, 256, 256) |
| 1 | 2 | Conv3D + LeakyReLU | (3, 3, 3) | (1, 1) | - | (1, 12, 256, 256) |
| 1 | 1 | InputRescaling | - | - | scale: 0.01 | (1, 12, 256, 256) |

magnification between the focal plane of the objective and the sensor plane was 18.35X, resulting in an object space pixel size of 0.354 $\mu$m. To experimentally characterise the optical encoder (point spread function), an artificial point source was generated by focusing a collimated laser diode (532 nm, CPS532, Thorlabs) to a diffraction limited spot using a second, higher NA, microscope objective (60X 0.9 NA LUMPLFLN60XW, Olympus) facing the primary microscope objective lens. This artificial point source was displaced about the focal plane of the primary microscope objective in 5 $\mu$m steps over a total range of 250 $\mu$m. 100 images were acquired at each plane, averaged, and dark frame subtracted.

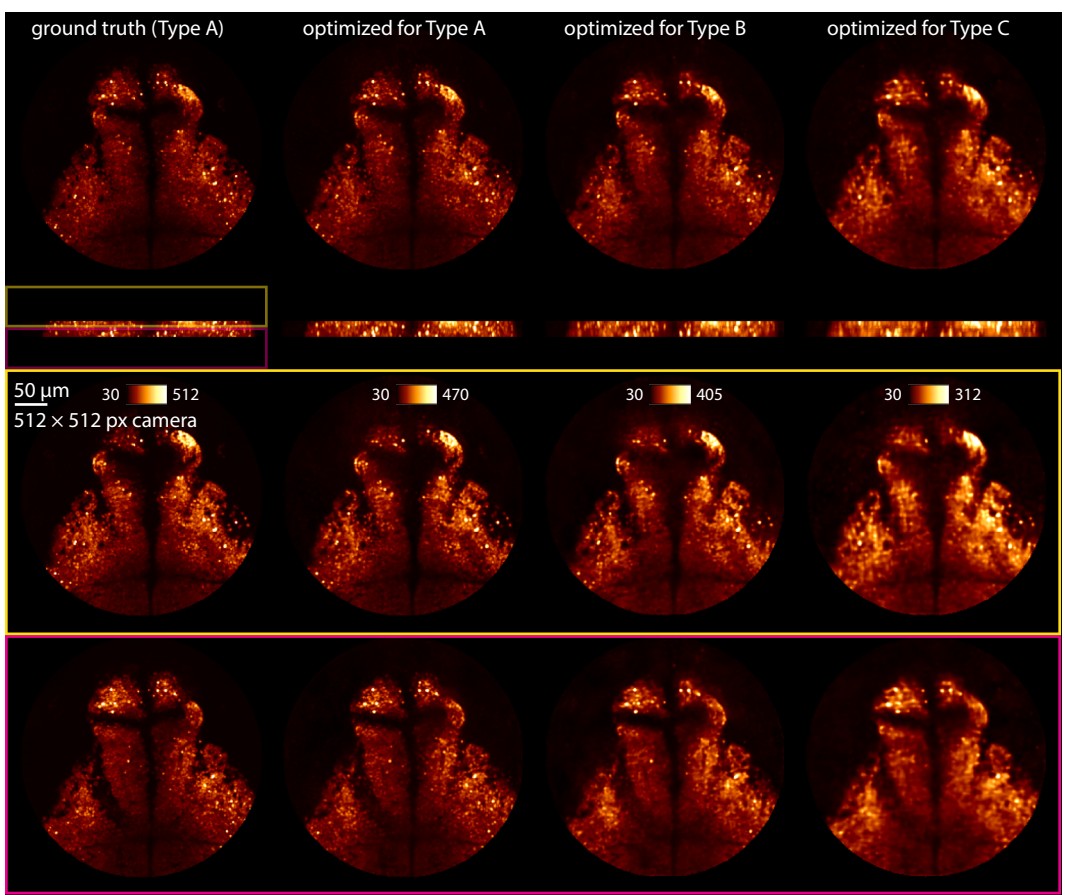

Figure 15: Slab views of an example Type A volume show that the phase mask optimized for Type A results in the best reconstructions. Note that the reconstruction with a phase mask optimized for Type A is almost identical to the ground truth, while the other phase masks create blurrier reconstructions. Slabs are xy max projections in thinner chunks as opposed to projecting through the entire volume. Colored boxes show which sample planes a particular slab comes from, corresponding to boxes in xz projection view at top.

Table 19: FourierNet2D detailed architecture (1 per plane)

| Layer type | Kernel size | Stride | Notes | Shape (C, D, H, W) |
|---|---|---|---|---|
| InputScaling | - | - | scale: 0.01 | (1, 1, 512, 512) |
| FourierConv2D | (512, 512) | (2, 2) | - | (5, 1, 512, 512) |
| LeakyReLU | - | - | slope: -0.01 | (5, 1, 512, 512) |
| BatchNorm2D | - | - | - | (5, 1, 512, 512) |
| Conv2D | (11, 11) | (1, 1) | - | (1, 1, 512, 512) |
| ReLU | - | - | - | (1, 1, 512, 512) |
| InputRescaling | - | - | scale: 0.01 | (1, 1, 512, 512) |

The phase mask (microscope parameters $\phi$) was optimized for this spatial light modulator (SLM) by choosing a number of pixels for the phase mask such that the desired pupil size would fit on the physical pixels of the SLM. In order to simulate high frequencies accurately, we upsample this phase mask to the number of pixels used for all our other simulations. This phase mask was optimized for Type B samples.

For visualization in Figure 5, we clipped any values of the measured optical encoder that were below 0 after dark frame subtraction to 0, then simulated imaging. We simulated the optical encoder using the same wavelength as the laser point source (532 nm) for the measured optical encoder. For both the

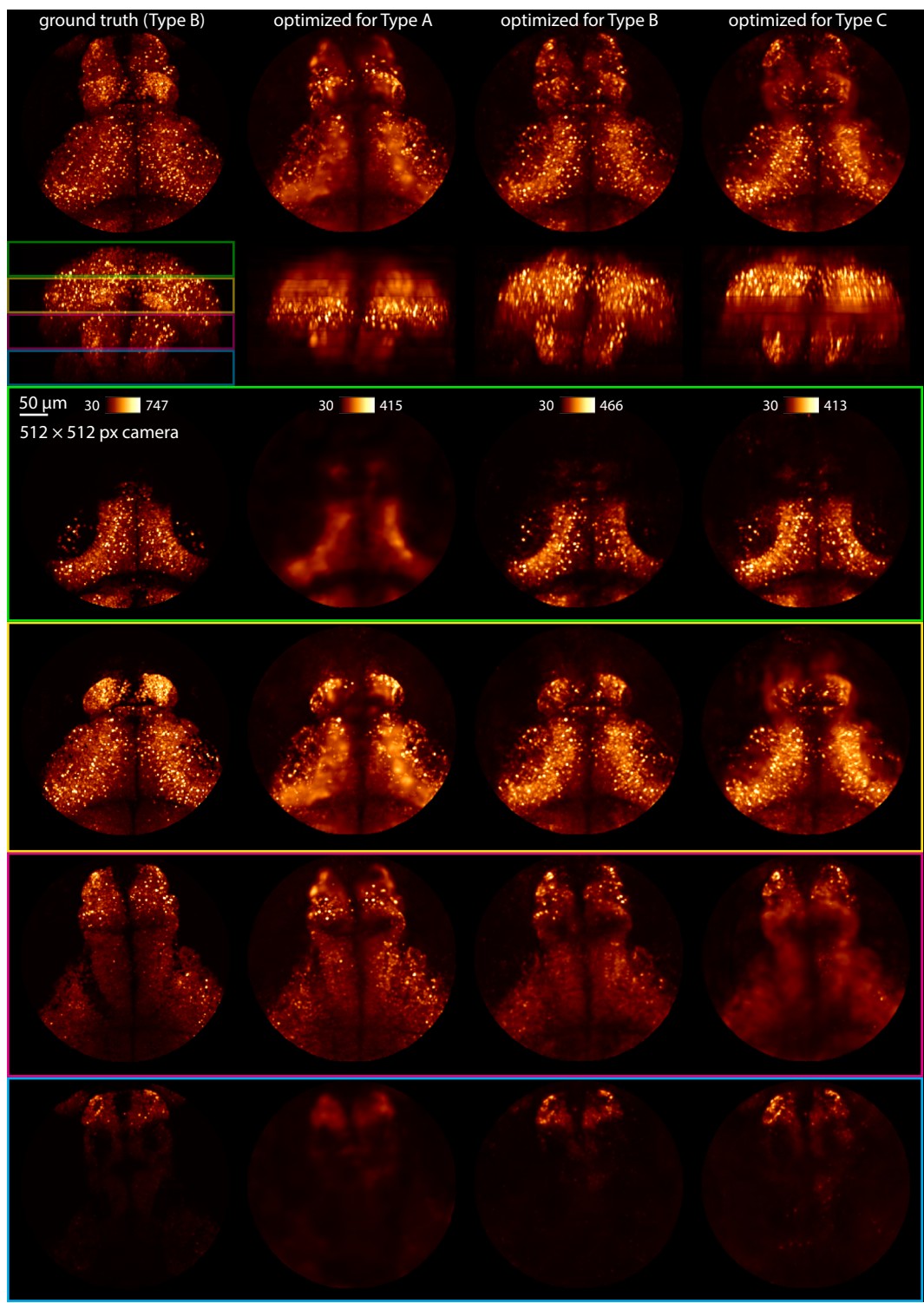

Figure 16: Slab views of an example Type B volume show that the phase mask optimized for Type B results in the best reconstructions; other phase masks result in blurrier reconstructions. Colored boxes show which sample planes a particular slab comes from, corresponding to boxes in xz projection view at top.

simulated and measured optical encoders, we scale the values to range from 0 to 1. We then simulated

Table 20: FourierNet3D detailed architecture (8 GPUs)

| Layer type | Kernel size | Stride | Notes | Shape (C, D, H, W) |
|---|---|---|---|---|
| InputScaling | - | - | scale: 0.01 | (1, 1, 512, 512) |
| FourierConv2D | (512, 512) | (2, 2) | - | (80, 1, 512, 512) |
| LeakyReLU | - | - | slope: -0.01 | (80, 1, 512, 512) |
| BatchNorm2D | - | - | - | (80, 1, 512, 512) |
| Reshape2D3D | - | - | - | (5, 16, 512, 512) |
| Conv3D | (11, 7, 7) | (1, 1, 1) | - | (5, 16, 512, 512) |
| LeakyReLU | - | - | slope: -0.01 | (5, 16, 512, 512) |
| BatchNorm3D | - | - | - | (5, 16, 512, 512) |
| Conv3D | (11, 7, 7) | (1, 1, 1) | - | (1, 16, 512, 512) |
| ReLU | - | - | - | (1, 16, 512, 512) |
| InputRescaling | - | - | scale: 0.01 | (1, 16, 512, 512) |

imaging using the same sample from our Type B dataset. The simulation for the measured optical encoder used a Type B sample interpolated to a resolution that matched the Orca-Flash camera.

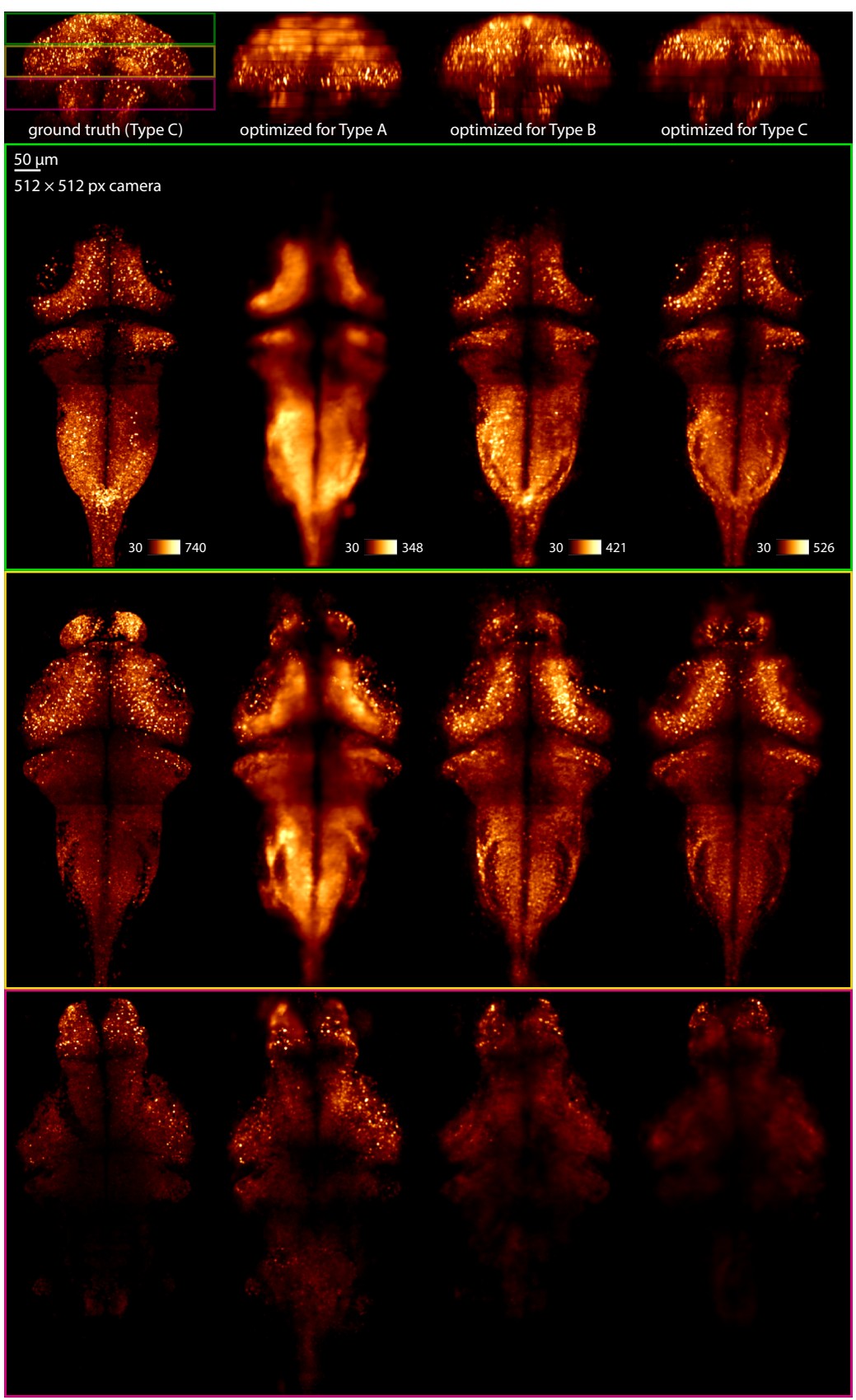

Figure 17: Slab views of an example Type C volume show that phase mask optimized for Type C provides most consistent reconstruction. Colored boxes have same meaning as Figures 15, 16.