# OpenReview forum: "FourierNets enable the design of highly non-local optical encoders for computational imaging"
_NeurIPS.cc/2022/Conference — NeurIPS 2022 Accept_

### Official Review · Reviewer_qEpE · 2022-07-11

**Rating:** 7
**Confidence:** 3
**Soundness:** 4 excellent
**Presentation:** 4 excellent
**Contribution:** 3 good

**Summary:**

This paper has two main contributions: (1) development of a parallelized differentiable wave optical encoder simulator and (2) a Fourier-space neural network which enables non-local optical decoding and outperforms previous deep learning-based image reconstruction strategies. Taken together, these two contributions enable end-to-end optimization of non-local optical encoding/decoding systems. The paper demonstrates both of these techniques on a new dataset of 3D snapshot zebrafish larvae microscopy images and demonstrates the reconstruction strategy on a previously published dataset for DiffuserCam lensless photography. The paper also demonstrates that the learned optical encoders can be implemented in practice on a prototype programmable microscope.

**Questions:**

The “weaknesses” section is structured as a list of questions I would like answered/clarified. To summarize:
- Do FourierNets with U-Net style pooling still outperform U-Nets?
- Is the U-Net in all experiments designed to have a receptive field that covers the entire image?
- Why are results reported with MSE and MSE+LPIPS loss for FourierNets in Table 1, but only for MSE+LPIPS for the other methods?
- Is the 2-step training procedure used for the U-Nets as well as the FourierNets?
- Is the loss function in Equation (4) used to train all methods in Table 2, including the U-Nets?

**Limitations:**

Yes, the authors have clearly addressed the limitations and the societal impact in Section 4. To summarize, the main limitation identified by the authors is that all of the results in this paper are presented on simulated data.

**Strengths And Weaknesses:**

[Note for AC: I don’t have direct experience with 3D snapshot microscopy or differentiable optical simulation — while I read through those details and they seem solid to me, the other reviewers may be more qualified to comment on that component of this work. My experience is more closely related to Fourier-space neural network architectures part of this work.]

# Strengths
- This paper is extremely clearly written — it is quite easy to follow, even as someone relatively unfamiliar with 3D snapshot microscopy.
- The experimental results are compelling and show clear improvements from the FourierNet architecture, both qualitatively and quantitatively.
- The appendix contains extensive, easy-to-follow characterization of the computation time and memory requirements of all methods studied, as well as detailed pseudocode and implementation details for the various algorithms described in the paper.

# Weaknesses
 I cannot find any major technical weaknesses with this paper and have only minor comments/questions/clarification requests here.
- There are currently two differences between FourierNets and standard ConvNets: The global-width convolution filters, implemented in Fourier space, and the different pooling strategy which crops in Fourier space, instead of, say, max or average pooling. If time/resources are available, a small ablation which compares FourierNets to FourierNets with image-space max/average pooling would further strengthen the argument that it is the global filter extent which is leading to performance gains. However, I don’t think this is strictly necessary for acceptance, because both of the implemented differences rely on the Fourier implementation of the networks used in this paper.
- Is the U-Net in all experiments designed to have a receptive field that covers the entire image? If not, this comparison should be made to show that, even if the U-net has such a wide receptive field, the local operations at each layer are ill-suited to tasks with a wide point spread function. If so, it may be worth it to clarify this explicitly in the text.
- Why are results reported with MSE and MSE+LPIPS loss for FourierNets in Table 1, but only for MSE+LPIPS for the other methods? Do the other methods do worse with MSE loss?
- Lines 219-223 describe a two-step training process — was this used for generating the metrics in Table 2 just when training FourierNets as microscopes? Or also when using the UNets?
- Is the loss function in Equation (4) used to train all methods in Table 2, including the U-Nets? I assume so, since no other loss functions are mentioned, but if not, ablations are needed to show that the architecture, and not the loss function, are contributing to the improvement observed here.
- There is limited technical novelty _from an ML perspective_ in this work (neural networks operating via elementwise operations in the Fourier space have previously been developed, as cited in the manuscript). That said, I do think that the use of FourierNets for end-to-end optimization of the types of imaging systems in this paper is novel and a solid computational imaging advance.
- Very minor corrections: Figure 2 caption’s third sentence should read “Bottom row.” And I believe Line 235 should reference Table 2.

---

> ### Author Response · Authors · 2022-08-02
> **Response to Reviewer qEpE**
>
> Thank you for your thoughtful review. We have addressed your list of questions in order:
>
> 1. **Image-space pooling**: This is an interesting suggestion. We did not investigate pooling in image space versus Fourier space. We agree that this constitutes an interesting architecture fusing FourierNets and UNets in a conceptually different manner, a potential future direction.
>
> 2. **UNet receptive field**: Our UNets are indeed designed to have a global receptive field. UNet2D has 8 levels and a receptive field of 4,539 x 4,539 pixels while UNet3D has 4 levels and a receptive field of 279 x 279 pixels, both of which are larger than the 256 x 256 pixels of the input camera image. We appreciate your suggestion and have edited our text and supplement to make this explicit.
>
> 3. **MSE+LPIPS vs. MSE only**: The other methods in Table 1 were not trained by us and come directly from Monakhova et al. 2019 [25] where they did not report the results from MSE only training. They do mention that their results were not as good when trained without LPIPS. We found that training with MSE was sufficient for our method. We have clarified this point in our text.
>
> 4. **Table 2 two-stage trainings**: All the 3D snapshot microscopy results are using the two-stage process (both FourierNet and UNet). Training is a two stage process in which a planewise reconstruction network with fewer parameters and 2D convolutions is used during optimization of the optical encoder parameters, then once the optical encoder parameters are fixed a more powerful reconstruction network with 3D convolutions is trained. We show the type of network used for the first stage to train the optical encoder in the first column, and the type of network used in the second stage for training the reconstruction network only (given a fixed optical encoder) in the second column. We have clarified this point in our text.
>
>  5. **Loss function**: We do indeed train all our snapshot microscopy networks using the same loss function described in (4). We have clarified this point in our text.
>
> We appreciate your additional corrections. We have updated the caption in Figure 1 and line 235.

---

> > ### Comment · Reviewer_qEpE · 2022-08-08
> > **Thanks for the response!**
> >
> > Thanks to the authors for their point-by-point response to my questions; these responses confirm that the small points I raised are not issues for this paper. I will keep my previous score of 7 (Accept) for this paper.

---

### Official Review · Reviewer_FwUp · 2022-07-11

**Rating:** 7
**Confidence:** 4
**Soundness:** 3 good
**Presentation:** 3 good
**Contribution:** 3 good

**Summary:**

The paper proposes an end-to-end (E2E) optimized computational imaging method to reconstruct 3D volumes (or 2D images). The method uses simulated optical encoders which encode the input into a 2D encoded phase image which is then decoded by a Fourier transform-based neural network. The method is then evaluated on 3D snapshot microscopy data where the input is a 3D volume, and DiffuserCam-based lensless photography, where the input is the intermediate 2D phase image itself. The method outperforms existing methods and will be openly be made (which the reviewer acknowledges very positively).

The author’s claims are:
1.	A multi-GPU fast implementation of the simulated optical encoder where the parameterized phase mask can be optimized E2E
2.	Fourier Transform-based decoder converts the intermediate input into Fourier space and does convolution (actually element-wise multiplications) that achieves faster computation of the final output. This helps the network to reduce the locality bias and learn from a wider global context.
3.	They collected extra data to show the merit of their method in a region-of-interest (ROI) specific 3D volume imaging case.
4.	SOTA results on the following tasks – 3D snapshot microscopy and DiffuserCam-based lensless photography
5.	This is the first time that a direct E2E optimization of the spatial light modulator (SLM) based optical encoders have been proposed. They back this claim using the simulation result and a single case of implementation on a programmable microscope.


**Questions:**

* A brief explanation of the non-local optical encoder would have been helpful.
* Are there other baselines you could compare your approach agains?

**Limitations:**

I believe so.

**Strengths And Weaknesses:**

Strengths:
1.	The paper addresses an important area in optics and clearly defines the existing limitations of exploiting the latest advances in end-to-end machine learning methods and optimizing the optical encoding system.
2.	The arguments, concepts, assumptions, and methods are lucidly explained by the authors in the paper.
3.	The paper builds on the advances in multi-GPU-based parallelism and efficient deployment of neural networks for a problem that is otherwise not feasible due to limitations in computing.
4.	The author’s assumption that the locality bias by having a small filter size in existing methods cannot effectively capture then information from the image formation process, which is non-local, sounds plausible. To address this problem, the authors propose a method using Fourier transform and using larger filter sizes.
5.	Relevant papers have been cited and tricky parts have been described verbose in the appendix.
6.	The overview figure clearly depicts the proposed method.
7.	The use of Fourier domain convolutions reduces the computation time and increases inference speed.
8.	The loss function has been designed to give more importance to the high frequencies of the input which are more important than low frequencies which are relatively less important.
9.	The results on DiffuserCam dataset corroborate the working of the decoder in capturing the high frequencies better than existing methods.
10.	The authors show that their method can indeed be implemented on a programmable microscope as a prototype.
11.	The results on the 3D microscopy dataset show that:
a.	The E2E optimization outperforms non-E2E methods in reconstruction quality with fewer parameters.
b.	The proposed method can be optimized for ROI-specific optimizations, that achieve better results than existing methods.

Weaknesses:
1.	The application of this method seems to be in a very limited field which is a differentiable simulation of optical encoders.
2.	The authors could have shown the result on 1~2 more datasets.
3.	UNets have been there for a while. Are they indeed the best baseline method to compare the presented method against?
4.	There is no theory or citations to back the claim made in line 136.
5.	An entire multi-GPU setup is required for the optimizations in the proposed method, which makes it not very accessible for many potential users.

---

> ### Author Response · Authors · 2022-08-02
> **Response to Reviewer FwUp**
>
> Thank you for your positive and detailed review. We appreciate your advice about clarity and have updated our introduction to briefly describe what we mean by an optical encoder. Optical encoders can usually be modeled as linear transfer functions implemented by the optics. In special cases, such as for the optical systems explored in this paper, they can be represented as a linear convolution filter called a “point spread function” which has its coefficients computed by a wave optics simulation, dependent on the physical parameters of the optical system. We provide a more detailed description in the supplement (A.1). We hope our views on some of the additional points you raise are helpful:
>
> 1. **Limited field**: The NeurIPS call for papers specifically includes “applications”, “machine learning for sciences”, and “neuroscience”. Our paper describes an exciting project at the intersection of (deep) (model-based) machine learning, differentiable physical simulations, computational optics, and neuroscience. We believe that ML-powered imaging is an important emerging research application in the ML community and believe it will have a broad and diverse readership.
>
> 2. **More datasets**: We applied our method to two different datasets (DiffuserCam, zebrafish) and two different problems (lensless photograph reconstruction, 3D snapshot microscopy). Further, our 3D snapshot microscopy task required the creation of a new dataset at great expense. We created a large number of transgenic zebrafish and performed confocal imaging of their entire nervous system at high resolution using confocal microscopy. We will release this new dataset to the community upon acceptance. Most papers in computational optics contribute or evaluate on only 1 to 2 datasets (e.g. Monakhova et al. [25], Ikoma et al. [3]).
>
> 3. **Baselines**: We focus on end-to-end optimization of optical encoders in this work, and to the best of our knowledge, most previous methods for this style of optimization have used various forms of the UNet [1 - 7], hence why we focus on UNet comparisons. These works range from April 2019 to December 2021, so they are quite recent. Our experiments on DiffuserCam do compare against non-UNet baselines, which involve differentiable deconvolution methods based on unrolled iterative optimization that use UNets for denoising. In response to reviewer wkck, we have also implemented the architechture used by Ikoma et al. [3], and are currently training end-to-end optical encoder optimization using this method. While the training is still in progress, early results suggest that the performance of this architecture is substantially worse than our method, but indeed better than a vanilla UNet. We anticipate completion of the training in a few days and will update the manuscript with a comparison with the Ikoma et al. [3] network.
>
> 4. **Locality bias**: We only intended to hypothesize that UNets have a locality bias. We have reworded to make this clear: “believe” -> “speculate”. While it might be possible to address this claim more theoretically, this is not the point of our paper. All we can demonstrate is that the UNet under-performs the FourierNet when it comes to the engineering of optical encoders, and that it consistently leads to sub-optimal optical encoders which are local rather than non-local.
>
> 5. **Multi-GPU**: Our simulation uses multiple GPUs to enable gradient based optimization in a reasonable amount of time. The support of the convolutional non-local optical encoder is determined by the physical parameters of the system, and for snapshot microscopy the physics-based optical simulation becomes quite large. Our parallel multi-GPU differentiable optical simulation framework is therefore one of the major contributions of this work.

---

### Official Review · Reviewer_wkcK · 2022-07-11

**Rating:** 6
**Confidence:** 5
**Soundness:** 4 excellent
**Presentation:** 4 excellent
**Contribution:** 3 good

**Summary:**

This paper proposes using a shallow network with a large convolutional kernel for end-to-end (E2E) learning of computational imaging systems. The proposed large convolutional kernel (used only in the first layer) is implemented in the Fourier domain. The proposed network is used to reconstruct experimentally captured diffuserCam data, where it outperforms SoTA methods. It is also used to E2E design a snapshot 3D microscope. This new design significantly outperforms a U-net based system (in simulation), though it is not compared with lenslet based systems. The authors wrote and plan to release parallelized software which allows one to simulate wave propagation using 8 GPUs.

**Questions:**

How do E2E designed masks compare to lenselet based masks?

Could the authors speak to why an approximate inverse (i.e., A^dagger y or Wiener deconv) before a U-net doesn't adequately model non-local effects.

**Limitations:**

Addressed

**Strengths And Weaknesses:**

# Strengths

Using a large receptive field in when dealing with optical systems with large PSFs is well motivated and the proposed design seems to work better than Unets.

Proposed method produces SOTA results on diffusercam data

Paper is well-written and easy to read

Proposed methodology could lead to interesting new microscope designs

The multi-GPU simulator could be useful for a number of research groups

# Weaknesses

The paper doesn't adequately discuss or compare against methods (particularly in the 3D experiments) methods that first approximately inverse the optical system before applying a NNs [3,A]. Learned Wiener filters [A] seem quite similar to the proposed technique.

No experimental 3D microscopy results (with the proposed optics).

No comparisons with classical or lenslet based approaches to snapshot 3D imaging.



[A]Yanny, Kyrollos, et al. "Deep learning for fast spatially varying deconvolution." Optica 9.1 (2022): 96-99.

---

> ### Author Response · Authors · 2022-08-02
> **Response to Reviewer wkck**
>
> Thank you for your thoughtful review. We have focused in this work on enabling end-to-end highly non-local PSF engineering for the snapshot microscopy domain, where end-to-end optimization at the phase mask pixel level has never been performed, and at a scale that has not been previously possible.
>
> **How well does an approximate inverse before a U-net work?**
>
> Due to the focus of our paper, we have mainly compared against methods that have been used for PSF engineering, hence our comparisons against the UNet. We do appreciate your references and are working on experiments with a UNet+Wiener filter approximate inverse method for snapshot microscopy. As you point out, Ikoma et al. [3] did in fact use a Wiener filter approximate inverse as an input to a UNet for PSF engineering. In response to your review, we have implemented this architecture and are currently training end-to-end PSF optimization using this method. While the training is still in progress, early results suggest that the performance of this architecture is substantially worse than our method, but indeed better than a vanilla UNet. We anticipate completion of the training in a few days and will update the manuscript with a comparison with the Ikoma et al. [3] network. Additionally, we have included a reference to the similar, more recently published, Yanny et al. [A] paper, and we thank you for pointing out this reference.
>
> **Comparison to lenslet based masks**
>
> In our paper, we focused on end-to-end phase mask optimization at the pixel level. In contrast, past work for snapshot microscopy did not involve end-to-end optimization, and considered reduced dimensional parametrizations of the phase mask, as arrays of lenslets , e.g. from Miniscope 3D [19]. It is indeed striking that our optimized point spread function produces pencil-like elements, similar to [9, 10, 19]. As you can see from the supplement, our optimized phase masks (Figures 10 and 11 in the supplement) are very high frequency and look qualitatively different from lenslet-based phase masks. We speculate that our phase masks make more efficient use of the pupil in directing light to individual pencils from different parts of the pupil. Based on your review, we have updated our discussion to mention these  qualitative differences.
>
> **Experimental results with proposed optics**
>
> We should point out that while we do not yet have a full 3D snapshot microscopy pipeline using the proposed optics, we do show preliminary measurements of the point spread function implemented on the proposed optical system (Fig 5), and there is good agreement between simulation and experiment. This proves that our engineered optical encoder can indeed be implemented on our designed hardware, and that we have a faithful simulation of our optical system. Since our main claim is purely computational, and is only about our ability to **successfully optimize non-local optical encoders** and not about the hardware, we believe that we have fully demonstrated our claims.

---

> > ### Comment · Reviewer_wkcK · 2022-08-09
> > **Thanks**
> >
> > Thanks for the response. Adding results that use an approximate inverse to start will better contextualize the results and improve the submission.

---

> > > ### Author Response · Authors · 2022-08-09
> > > **New results added, paper updated**
> > >
> > > As you might have already noticed, the paper has already been updated with these new results. We hope you will consider updating your score.

---

### Author Response · Authors · 2022-08-08
**Response to all reviewers and summary of changes**

We thank the reviewers for their thoughtful reviews. We note that our paper was reviewed favorably, and all three reviewers recommended acceptance. We have responded to all the feedback, performed additional experiments during the feedback period to add a comparison to another baseline method, and updated our paper to address all the feedback. We hope the reviewers will find that our changes have resulted in a stronger paper, and update their scores accordingly.

Here is a summary of our responses and changes:

* Our work focuses on enabling end-to-end, pixelwise optimization of non-local optical encoders for snapshot microscopy, which has never been done before. Due to this focus, we compare mainly against the UNet, which is the main network architecture used for end-to-end optical encoder optimization in recent works from April 2019 to December 2021 [1 - 7].
* In response to reviewer wkck, we have implemented as an additional baseline method the Wiener deconvolution + UNet method as in Ikoma et al. [3]. We find that it improves the reconstruction performance over just a UNet, but still does not perform as consistently as our FourierNet architecture (Figure 3, Table 2). The resulting optical encoder also does not make full use of the 2D camera image as compared to our FourierNet (Figure 3, Appendix Figure 8).
* We have included a citation to Yanny et al. 2022 [27] as suggested by reviewer wkck.
* As suggested by reviewer wkck, we have included discussion about our resulting phase masks (optical encoders), which appear qualitatively different from more traditional lenslet-based phase masks. We speculate that our phase masks make more efficient use of the pupil in directing light to individual pencils from different parts of the pupil.
* As suggested by reviewer FwUp, we updated our introduction with a brief description of optical encoders.
* As suggested by reviewer FwUp, we have clarified that we are only speculating that UNets have a locality bias based on our experimental results, which show that the UNet consistently leads to sub-optimal, local optical encoders compared to the FourierNet. While it might be possible to address this claim more theoretically, this is not the point of our paper.
* As suggested by reviewer qEpE, we have clarified in our main text that our UNet2D and UNet3D architectures do indeed have global receptive fields and included these details in the supplement.
* As suggested by reviewer qEpE, we have clarified that results in Table 1 are taken from Monakhova et al. 2019 [25] (not trained by us) and do not include training with MSE alone because the authors only reported MSE + LPIPs. The authors claimed that MSE+LIPIPS out-performed MSE alone for their experiments.
* As suggested by reviewer qEpE, we have clarified which experiments are trained in two stages in Table 2.
* As suggested by reviewer qEpE, we have clarified that our loss in equation (4) is used for all snapshot microscopy trainings.
* The metrics for snapshot microscopy in Table 2 are now computed ignoring empty (zero) regions in the original sample. The metrics reported in the rest of the paper already used this convention (Table 3). We apologize for the earlier inconsistency.
* We have updated timing results for forward passes (Table 1, Table 2), all using TITAN Xp GPUs for consistency. Relative speeds have not changed.

---

### Meta-Review · Area_Chair_QzWf · 2022-08-25

**Recommendation:** Accept
**Confidence:** Certain

**Metareview:**

This paper develops a multi-GPU differentiable simulation of a programmable microscope, wherein a large convolutional kernel used in the first layer is implemented in the Fourier domain.  The work outperforms sota approaches for lensless photography, and it allows the end-to-end design of a snapshot 3D microscope which beats state of the art systems in simulation.  Overall the work provides a compelling demonstration of the power of recent technical advances in ML to lead to demonstrable improvements (in simulation) of computational imaging methods.

**Award:**

No

---

### Decision · Program_Chairs · 2022-09-14

Accept